# Describe, Explain, Plan and Select:
# Interactive Planning with Large Language Models
# Enables Open-World Multi-Task Agents

**Zihao Wang**[1,2], **Shaofei Cai**[1,2], **Guanzhou Chen**[3], **Anji Liu**[4], **Xiaojian Ma**[4], **Yitao Liang**[1,5]*
**Team CraftJarvis**
[1]Institute for Artificial Intelligence, Peking University
[2]School of Intelligence Science and Technology, Peking University
[3]School of Computer Science, Beijing University of Posts and Telecommunications
[4]Computer Science Department, University of California, Los Angeles
[5]Beijing Institute for General Artificial Intelligence (BIGAI)
{zhwang,caishaofei}@stu.pku.edu.cn,rayment@bupt.edu.cn
liuanji@cs.ucla.edu,xiaojian.ma@ucla.edu,yitaol@pku.edu.cn

## Abstract

We investigate the challenge of task planning for multi-task embodied agents in open-world environments.[2] Two main difficulties are identified: 1) executing plans in an open-world environment (e.g., Minecraft) necessitates accurate and multi-step reasoning due to the long-term nature of tasks, and 2) as vanilla planners do not consider how easy the current agent can achieve a given sub-task when ordering parallel sub-goals within a complicated plan, the resulting plan could be inefficient or even infeasible. To this end, we propose "Describe, Explain, Plan and Select" (**DEPS**), an interactive planning approach based on Large Language Models (LLMs). DEPS facilitates better error correction on initial LLM-generated *plan* by integrating *description* of the plan execution process and providing self-*explanation* of feedback when encountering failures during the extended planning phases. Furthermore, it includes a goal *selector*, which is a trainable module that ranks parallel candidate sub-goals based on the estimated steps of completion, consequently refining the initial plan. Our experiments mark the milestone of the first zero-shot multi-task agent that can robustly accomplish 70+ Minecraft tasks and nearly double the overall performances. Further testing reveals our method's general effectiveness in popularly adopted non-open-ended domains as well (i.e., ALFWorld and tabletop manipulation). The ablation and exploratory studies detail how our design beats the counterparts and provide a promising update on the `ObtainDiamond` grand challenge with our approach. The code is released at https://github.com/CraftJarvis/MC-Planner.

## 1 Introduction

Developing multi-task agents that can accomplish a vast and diverse suite of tasks in complex domains has been viewed as one of the key milestones towards generally capable artificial intelligence [36, 1, 5, 10, 25]. To enable such capabilities, earlier works have suggested employing a hierarchical goal execution architecture [2, 4], where a planner generates action plans that would then be executed by low-level goal-conditioned controllers. This architecture has been delivering promising progress in

---

*Corresponding Author.

[2]We borrow the term "open world" from the game community. It highlights that the agent can navigate inside a diverse environment and accomplish open-ended tasks freely.

37th Conference on Neural Information Processing Systems (NeurIPS 2023).

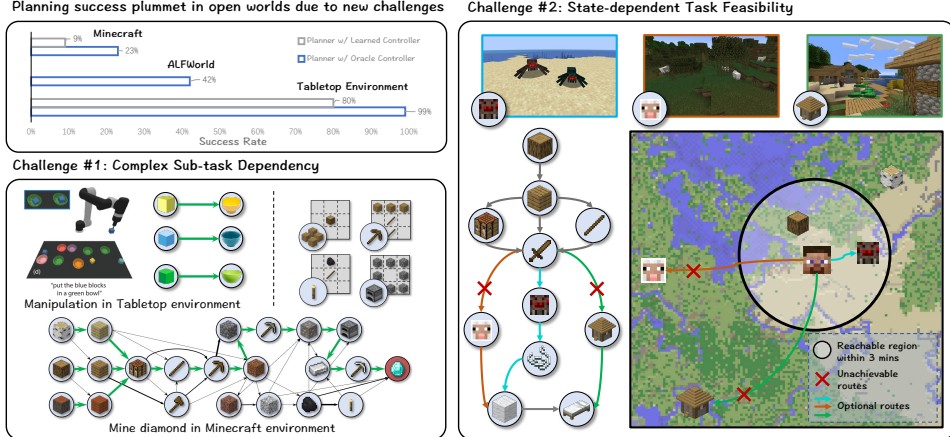

Figure 1: **Planning success rates plummet in open worlds due to new challenges**.

many robotics domains, including table-top and mobile manipulation [46, 4], 2D shape drawing [20] and table rearrangement [17]. However, whether such success can be transferred to a more open-ended world with unlimited exploration areas and internet-scale knowledge remains open [14, 10, 13, 12, 19].

To understand the gap, we run Inner Monologue [17], a general and competitive hierarchical goal execution model on a typical open-world domain Minecraft [18, 14, 10] and two classical robotic environments ALFWorld [41] and Tabletop environments [40, 4]. The algorithm uses a Large Language Model (LLM) based planner that contains domain-specific knowledge for all three environments. In all environments, we use either an Oracle goal-conditioned controller or a learned one. Results are shown in the bar plot in Figure 1. First, even when the Oracle controller is used, the success rate of executing Minecraft tasks is much less than that of the other environments. Next, the task failure rate becomes even higher in Minecraft when the learned controller is substituted. Both failures originate from unique challenges brought by open-world environments, which we identify in the following.

First, compared to canonical environments (e.g., Atari [29] and robotic control suite [40]), open worlds have highly abundant object types with complex dependency and relation. As a result, ground-truth plans typically involve a long sequence of sub-goals with strict dependencies. As Figure 1 challenge #1 suggests, it requires at least 13 sub-goals executed in proper order to obtain a diamond in Minecraft, while in Tabletop a task is typically no more than a few consecutive sub-goals.

Another challenge brought by the complicated tasks in an open-ended world is the feasibility of the produced plans. Consider the example shown in Figure 1 (challenge #2). To craft a bed in Minecraft, the fastest way is by either slaughtering a sheep to obtain wool, which can be used to craft beds, or collecting beds from a village. However, since no sheep or village is reachable by the agent within 3 minutes of gameplay, to craft a bed efficiently, the agent should choose to slaughter a spider and use materials (e.g., string) it drops to craft wool, and then a bed. That is, when dealing with a task that can be completed by executing multiple possible sequences of sub-goals, the planner should be able to select the best route based on the current state of the agent. However, the complex and diverse state distribution of open-world environments makes state awareness hard to achieve.

To tackle these problems, we propose "Describe, Explain, Plan and Select" (**DEPS**), an interactive planning approach based on Large Language Models (LLMs) to alleviate the aforementioned issues. The key to tackling the first challenge is to effectively adjust the generated plan upon failure. Specifically, whenever the controller fails to complete a sub-goal, a *descriptor* will summarize the current situation as text and send it back to the LLM-based planner. We then prompt the LLM as an *explainer* to locate the errors in the previous plan. Finally, a *planner* will refine the plan using information from the descriptor and explainer. To improve the feasibility of generated plans conditioned on the current state, which is the second identified challenge, we use a learned goal-*selector* to choose the most accessible sub-task based on the proximity to each candidate sub-goal.

Our experiments are conducted on 71 tasks in open-ended Minecraft without any demonstration. Given the goal-conditioned controller for atom sub-tasks (i.e., mine log and mine stone), our zero-

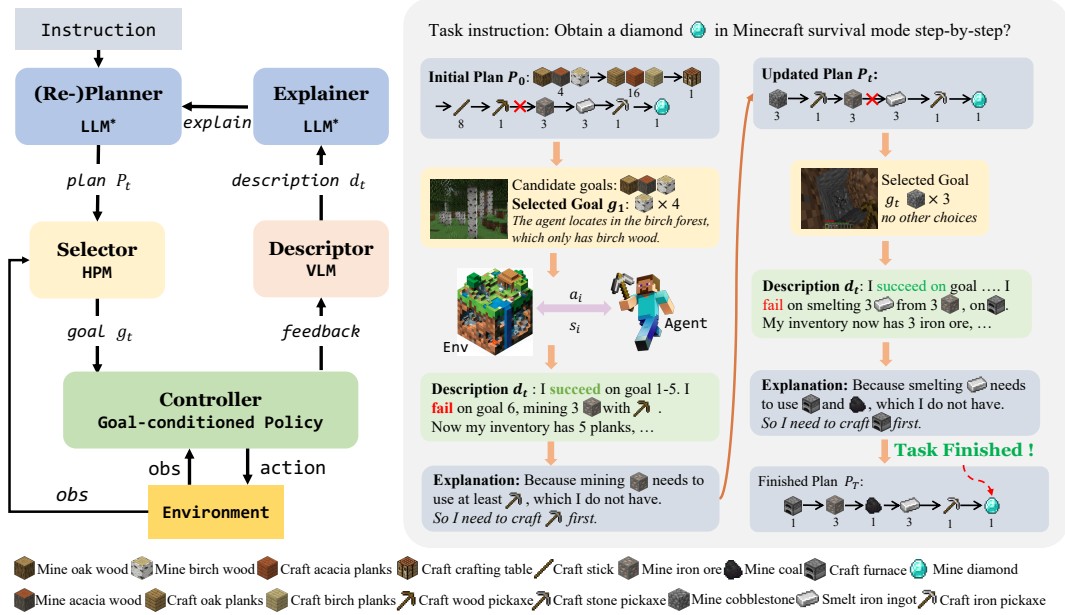

Figure 2: **Overview of our proposed interactive planner architecture**.

shot[3] LLM-based planner can finish all tasks within a limited horizon (3000-12000 steps for different tasks). We find DEPS outperforms all language planner baselines by nearly doubling the overall success rate, with the same initial state and goal-conditioned controller. Our ablation and exploratory studies then explain how our approach beats the counterparts and becomes the first planning-based agent that accomplishes the challenging ObtainDiamond task. DEPS does not require any planning training for the environment. Additionally, DEPS achieves between on-par and more than 50% relative improvement over existing or concurrent LLM-based planning methods on non-open-ended robotics domains such as ALFWorld [41] and Tabletop environments [40].

## 2 Background

We aim to develop an agent capable of solving long-horizon goal-reaching tasks using image observations and language goals. To accomplish this, we propose a combined approach involving goal-conditioned policies (termed controllers) and a planner. The goal-conditioned policies are trained to complete sub-goals, while the planner decomposes long-horizon tasks into a series of $K$ short-horizon sub-goals, $g_1, \ldots, g_K$, to be executed by the controller. At each time step $t$, the goal-conditioned policy $\pi(a_t \mid s_t, g_k)$ generates an action $a_t$ based on the current state $s_t$ and the specified sub-goal $g_k$.

**Planning with Large Language Models** Previous works have shown that LLMs such as Instruct-GPT [32] and Codex [8] can be used as zero-shot planners to generate sub-goal sequences for various tasks in embodied environments [16, 42]. Formally, given the task description $T$ as prompt $p$, LLM acts as a planner to decode $T$ into $K$ sub-goals, $g_1, \ldots, g_K$, which are then executed one by one by the low-level controller $\pi(a_t \mid s_t, g_k)$ to accomplish the task.

However, the above pipeline suffers from both challenges identified in Section 1. Regarding the first challenge, the probability of generating a flawless plan directly from the task description decreases significantly as the required number of sub-goals increases. Moreover, even when the LLM generates a correct plan, it is very likely that the plan is highly inefficient given the agent's current state (challenge #2). Prior works mostly focus on solving the first challenge by providing environmental feedback to the LLM through affordance functions [4], success detector [20] or scene descriptor [17]. However, although these approaches work well on many non-open-ended domains, they still suffer from high failure rates in open-world environments.

---

[3]Similar to [5, 16], "zero-shot" here means no gradient updates are performed. However we provide some related demonstrations as prompts during inference time.

# 3 Towards Reliable Planning in Embodied Open-World Environments

In this section, we first give an overview of our proposed interactive planning framework "Describe, Explain, Plan, and Select" (DEPS) for solving complex and long-horizon tasks in open-world environments (Sec. 3.1). Next, in Section 3.2, we elaborate how DEPS iteratively refines its plan to combat the first identified challenge. Section 3.3 introduces the *selector* module that is used to identify efficient plans in response to the second identified challenge.

## 3.1 DEPS Overview

As demonstrated in Figure 2, our agent (DEPS) consists of an event-triggered Descriptor, a Large Language Model (LLM) as Explainer and Planner, a goal Selector based on horizon prediction and a goal-conditioned controller. In the following, we use Minecraft as a running example to better elaborate our agent. Note that DEPS can be directly applied to other (non-)open-ended tasks.

We take a large language model (LLM) as a zero-shot *planner* of the agent to complete tasks. Given a goal command (e.g., `ObtainDiamond`) as task $T$, the LLM-based planner decomposes this high-level task into a sequence of sub-goals $\{g_1, \ldots, g_K\}$, as the initial plan $P_0$. The goals are instructions in natural language, such as `mine oak wood` (in Minecraft), find two cups (in ALFWorld), put block A on top of block B (in Tabletop Manipulation).

As described in Section 2, a controller is then invoked to execute the provided sub-goals sequentially through a goal-conditioned policy $\pi(a \mid s, g)$. However, the initial plan provided by the planner often contains errors, which results in execution failures of the controller. For example, the goal can not be finished only with a wooden pickaxe as shown in Figure 2. When failure pops up, the *descriptor* will summarize the current state $s_t$ and execution outcome of the most recent goal into text $d_t$ and send it to the LLM. The LLM will first try to locate the errors in the previous plan $P_{t-1}$ by *self-explanation*, e.g., the goal need to be executed with a stone pickaxe. Then it will re-plan the current task $T$ and generate a revised plan $P_t$ according to the explanation. In this process, the LLM is also treated as an *explainer* in addition to the *planner* role. The Descriptor, Explainer, and Planner will be detailed in Section 3.2.

$$
\begin{aligned}
\text{Description} &: d_t = f_{\text{DESC}}(s_{t-1}), \\
\text{Explanation} &: e_t = f_{\text{EX}}(d_t), \\
\text{Prompt} &: p_t = \text{CONCAT}(p_{t-1}, d_t, e_t), \\
\text{Plan} &: P_t = f_{\text{LM}}(p_t), \\
\text{Goal} &: g_t \sim f_{\text{S}}(P_t, s_{t-1}), \\
\text{Action} &: a_t \sim \pi(a_t \mid s_{t-1}, g_t)
\end{aligned}
\tag{1}
$$

As shown in Equation (1), DEPS will iteratively update the plan $P_t$ until the task is finished, where $f_{\text{DESC}}$ is the descriptor model, $f_{\text{LM}}$ denotes the language model as explainer and planner, $f_{\text{S}}$ is the selector model, $\pi$ is goal-conditioned policies from the controller.

To filter out inefficient plans, the *selector* is trained to predict the number of time steps remaining to achieve every goal $g_k$ in a set of parallel goals given the current state $s_t$. When the generated plan contains alternative routes, the selector uses this information to choose a suitable goal as the current goal $g_t$. For example, the horizon predicted by the selector of goal `acacia tree` is less than goal `oak tree` in `Savanna` biome, which leads to `chop acacia tree` as current goal $g_t$.

## 3.2 Describe, Explain and Plan with LLM Generates Executable Plans

Current LLM-based planners usually query the LLM once at the beginning of every episode and use the output plan throughout the episode [16, 42]. However, as demonstrated by Figure 1, such one-shot planning methods often fail on long-horizon tasks that require many sub-goals. This is caused by two major issues. First, since the correct plan for long-horizon tasks needs to respect various complex preconditions, it is extremely hard for the LLM to generate a flawless plan directly from the task instructions, resulting in failure when simply following the initial plan. Additionally, due to the unpredictable transition dynamics, some incidents may happen during the execution and make the initial plan non-executable. To remedy these problems, existing methods introduce feedback (e.g.,

**Prompt 1** Planner prompt template, Python-like code

```python
def craft_wooden_axe(initial_inventory={}):
    # step 1: mine 3 logs
    mine(obj = {"log":3}, tool = None)
    # step 2: craft 12 planks from 3 logs
    craft(obj = {"planks":12}, materials = {"log":3}, tool = None)
    # step 3: craft 4 sticks from 2 planks
    craft(obj = {"stick":4}, materials = {"planks":2}, tool = None)
    # step 4: craft 1 crafting_table from 4 planks
    craft(obj = {"crafting_table":1}, materials = {"planks":4}, tool = None)
    # step 5: craft 1 wooden_axe from 3 planks and 2 sticks on crafting table
    craft(obj = {"wooden_axe":1}, {"planks": 3, "stick": 2}, tool = "crafting_table")
    return "wooden_axe"
```

from success detector or scene descriptor) to reflect on the results of previous executions [17, 20, 4]. However, merely informing the LLM whether a sub-goal is completed is often insufficient to correct the planning error.

To remedy this, we propose "describe, explain and plan", a new interactive planning method to generate more executable and explainable plans. We start with rewriting the prompt into an interactive dialogue format as in ChatGPT [32] so that subsequent feedback can be passed to the LLM effectively. The produced plan is also augmented with the preconditions and effects of each goal. The structured prompt improves the readability and interpretability of the plan and facilitates error-locating when the execution fails later, as demonstrated in Prompt 1.

The *descriptor* will then collect the feedback generated by the agent during the execution of the task. The feedback can be practically obtained either by a person (human feedback [4]), or by a pre-trained vision-language model CLIP [35]. While the previous type of feedback needs intensive human involvement, the latter from the pre-trained model needs to be fine-tuned for the specific domain, which decreases the automation and generalization of the agent. On the contrary, Minecraft returns the 'info' and other high-level observations (such as biome, GPS, and compass), we can easily translate the unstructured information into structured language. Therefore we take the symbolic information available in the game and translate it into feedback description $d_t$ in this work. To avoid carrying unrelated information in the prompt, we further distill plan-related messages (e.g., inventory information, biome) as final event-level description $d_t$ as demonstrated in Figure 2.

Notably, we also treat the LLM as an *explainer* to explain why the previous plans $P_{t-1}$ failed. Specifically, by analyzing the current state from description $d_t$ and precondition of current goal $g_t$, the explainer can identify the reason why the current goal cannot be executed successfully. As shown in Figure 2, the reason may be *the current goal requires the use of an iron pickaxe, but the tool is not prepared in advance, or the current goal requires the use of 3 planks, but the currently available planks are not enough*. To implement this, we provide few-shot demonstrations to the LLM as in chain-of-thoughts prompting [45], as shown in Prompt 1. Finally, the LLM goes back to its role as a *planner* and re-plans the task with the explicit explanation of existing bugs in the previous plan $P_{t-1}$, ultimately generating an updated plan $P_t$ according to the explanation.

### 3.3 Horizon-Predictive Selector Yields Efficient Plans

Due to the abundance of objects and the compositional nature of their functionalities, there often exist multiple feasible plans to complete a task, i.e., there are usually multiple paths for the completion of a particular goal. However, despite the feasibility of all such plans, most of them are highly inefficient to execute in the current episode. For example, as shown in Figure 2, obtaining a `wood` can be done by chopping oak trees 🟫, birch trees 🪵, or acacia trees 🟥. But only oak trees are available in the `plains` biome. So the planner needs to choose oak trees since it is more efficient, as the agent does not need to travel to another biome.

On the other hand, there is no strict sequential requirement for some goals in the plan $P_t$, i.e., $g_i, g_j \sim P_t$ enjoy the same precondition, which means $g_i$ and $g_j$ can be executed in any order. As shown in Figure 1, the choice of different paths (sequences) may affect the execution efficiency of the plan $P_t$ as one goal might be closer to the agent. Always choosing the closer goal to execute first could yield more efficient plans and improve the final success rate under a limited episode length. Moreover, the dynamic nature of open-world environments further amplifies the impact of efficient

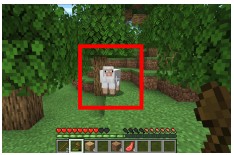 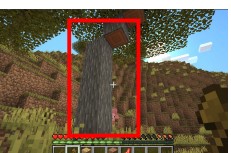 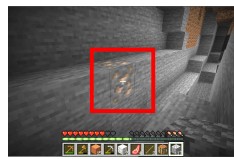 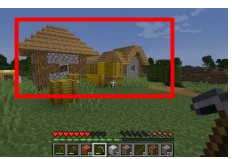

**Goal**: Meat*3
**Candidate Skill**: Kill Sheep *OR* Cow *OR* Pig
**Selection**: Kill Sheep
**Explanation**: Meet sheep first.

**Goal**: Log*2
**Candidate Skill**: Chop Oak *OR* Birch *OR* Acacia Tree
**Selection**: Chop Acacia Tree
**Explanation**: Savanna biome only has Acacia tree.

**Goal**: Coal*1 AND Iron_Ore*1
**Candidate Skill**: Mine Coal *AND* Iron_Ore
**Selection**: Mine Iron_Ore
**Explanation**: Meet iron_ore first.

**Goal**: Survive in Night.
**Candidate Skill**: Sleep in bed *OR* Dig down.
**Selection**: Sleep_in_bed
**Explanation**: Village has beds.

Figure 3: **Selection Demonstration from "Selector"**. Given parallel sub-goals, i.e. candidate skills, our Selector will determine the sequence in which to carry out these sub-goals based on their current proximity to the agent and modify the original plan produced by the LM planner.

plans on the success rate. For example, in Minecraft, if the agent chooses to execute a further goal like `collect wood` first, the much closer target `sheep` may disappear and be hard to find again.

In order to improve the efficiency of our plans, we propose to use a *selector* that selects the most efficient path with the highest execution success rate as the final plan. Specifically, we design a state-aware selector to choose the nearest goal under state $s_t$ as the current goal $g_t$ from the candidate goal sets in plan $P_t$. It predicts the goal distribution $p(g_t|s_t, P_t)$ under the current state $s_t$ and plan $P_t$, where $g_t \in G_t$, $G_t$ describes all current executable goals in $P_t$. A straight way to implement the selector is to leverage the semantic similarity between the current state and the goal text using a vision-language model (VLM) such as CLIP [35]. Nevertheless, this may not exactly reflect the difficulty of completing the goal since VLM lacks practical experience. For example, an "oak tree" in front of the agent could lead to high semantic similarity for the "chopping tree" goal, but it may be inefficient to achieve this goal if a canyon is in the middle between the agent and the oak tree.

To mitigate this, we implement a horizon-predictive selector that embeds practical task experience to accurately rank the goals based on their efficiency and feasibility. Here, we define the horizon of a goal $h_t(g) := T_g - t$ as the remaining time steps to complete the given goal, where $T_g$ is the time of completing goal $g$. This metric accurately reflects how quickly we can achieve the given goal from the current state. To estimate the horizon, we learn a neural network $\mu$ to fit the offline trajectories by minimizing the entropy loss $-\log \mu(h_t(g) \mid s_t, g)$, where $h_t$ is the ground-truth horizon in trajectories of completing goal $g$. Therefore, the goal distribution can be formulated as follows:

$$f(g_t \mid s_t, P_t) = \frac{\exp(-\mu(g_t, s_t))}{\sum_{g \in G_t} \exp(-\mu(g, s_t))}. \tag{2}$$

We set goal-sensitive Impala CNN [6] as the backbone of the selector. In practice, the horizon predictive selector can be jointly trained with the controller policies and share the backbone parameters [6].

## 4 Experiments

This section analyzes and evaluates our proposed "describe, explain, plan, and select" (DEPS) method. To minimize performance variation caused by the low-level controller, we standardize all experiments with one controller learned by behavior cloning. We refer to the details of this controller in Appendix C. In Section 4.1, we introduce our testing environments and our evaluation task set, consisting of the hardest 71 tasks from MCU SkillForgeChain [22]. In Section 4.2, we report our performance in the context of existing LLM-based planners. Ablation studies are conducted in Section 4.3. Finally, we pay close attention to the hardest task, `ObtainDiamond`, which is long-hailed as a major challenge in the community. The experiments on ALFWorld and Tabletop Manipulation environments are shown in Appendix A.

### 4.1 Experimental Setup

**Environment and Task Setting**   We first evaluate our proposed method in Minecraft, a popular open-world environment with both challenges discussed in Section 1. For better reflecting the performance of DEPS, we choose three Minecraft environments with different versions for better

evaluation, including Minedojo [10] with Minecraft 1.11.2, MineRL [3] with Minecraft 1.16.5, and MC-TextWorld [22] with Minecraft 1.19.2. Rules and items have something different in the above three Minecraft environments, which can better evaluate the dynamic and interactive planning abilities of DEPS.

Table 1: **Attributes of 8 meta tasks covering Task101**: We evaluate the algorithm on Minecraft Task101. We group the consisted 71 task into 8 different meta groups, with each focusing on testing a different aspect of our proposed method.

| Meta | Name | Number | Example Task | Max. Steps | Initial Inventory | Given Tool |
|------|------|--------|--------------|------------|-------------------|------------|
| MT1 | Basic | 14 | Make a wooden door. | 3000 | Empty | Axe |
| MT2 | Tool (Simple) | 12 | Make a stone pickaxe. | 3000 | Empty | Axe |
| MT3 | Hunt and Food | 7 | Cook the beef. | 6000 | Empty | Axe |
| MT4 | Dig-Down | 6 | Mine coal. | 3000 | Empty | Axe |
| MT5 | Equipment | 9 | Equip the leather helmet. | 6000 | Empty | Axe |
| MT6 | Tool (Complex) | 7 | Make shears and bucket. | 6000 | Empty | Axe |
| MT7 | IronStage | 13 | Obtain an iron sword. | 6000 | Empty | Axe |
| MT8 | Challenge | 1 | Obtain a diamond! | 12000 | Empty | Axe |

We choose 71 tasks from the Minecraft Universe Benchmark SkillForgeChain [22] for evaluation. These tasks are related to items that can be obtained in the Minecraft overworld. To better present the results, we divide the 71 Minecraft tasks into 8 meta groups according to the ingredients and function of the tasks, i.e., MT1-MT8. The instruction for every task is written in natural language, e.g., `make a wooden door` in MT1 (Basic group) and `obtain a diamond` in MT8 (Challenge group), as illustrated in Table 1. Considering how long it typically takes human players to complete each task as a ballpark [14], we set different maximum episode steps for different meta tasks from 3000 (for easiest **Basic** tasks) to 12000 (for the hardest **Challenge** tasks). The names, number of required skills, and functions of all tasks are listed in Appendix B. We give an empty inventory for every task in Survival mode and require the agent to obtain every item from the environment by itself. Note that our agent will be summoned in different environments randomly for each evaluation. Biomes and initial positions are also different each time. Following the previous work [18], we take the success rate as the evaluation metric.

**Baselines**   We compare DEPS with other language-based planners, including GPT as Zero-shot Planner(GPT) [16], ProgPrompt(PP) [42], Chain-of-Thought(CoT) [45], Inner Monologue(IM) [17], and Code as Policies(CaP) [20]. For all baseline models, we use the same demonstration example in the prompt, the same LM model from OpenAI, and the same controller in all tasks for a fair comparison. Since these methods were not originally experimented with Minecraft, we reproduce them to conform to the Minecraft specification based on prompt and feedback template design. All planner methods access the LLM model through OpenAI API (`text-davinci-03` model [32] for GPT, CoT, and IM, and `code-davinci-02` model [8] for PP, CaP, and Ours). All hyperparameters of LLM (including the *temperature* and *best_of*, etc.) are kept as default. We also list the full prompt of all different methods in Appendix G.

## 4.2   Main Results

Every task is executed 30 times and the average results in Minedojo [10] for every meta task are listed in Table 2. Our approach achieves the best performance with all meta tasks. As the complexity of the task increases from MT1-MT8, the planner usually needs to give more accurate task steps (i.e., longer goal sequence) to achieve the final task. Therefore the success rate of all agents decreases with the reasoning steps increasing. Starting from MT6, almost all existing LLM-based planners fail (nearly 0 success rate). DEP (w/o Selector) already consistently beats existing LLM-based planners in all meta tasks with a significant margin. This validates that "describe, explain and plan" can estimate the reason for current plan failure and correct the original flawed plans. Due to the limited maximum episode length and restricted control success rate for a hard goal (e.g., `Mine diamond with iron_pickaxe`), the final success rate is still capped.

Table 2: Success rates of DEPS and existing LLM planners on Minecraft Task101. The full task-by-task list is in Appendix F.

| Methods | MT1 | MT2 | MT3 | MT4 | MT5 | MT6 | MT7 | MT8 | AVG |
|---|---|---|---|---|---|---|---|---|---|
| GPT[16, 32] | 25.85±24.8 | 47.88±31.5 | 10.78±14.6 | 7.14±9.0 | 1.98±5.9 | 0.0±0.0 | 0.0±0.0 | 0.0±0.0 | 15.42 |
| PP[42] | 30.61±23.6 | 40.09±30.6 | 17.13±19.1 | 16.00±17.3 | 3.21±4.9 | 0.47±1.3 | 0.60±2.2 | 0.0±0.0 | 16.88 |
| CoT[45] | 40.24±30.8 | 55.21±26.8 | 6.82±11.6 | 4.76±8.2 | 1.73±5.2 | 0.0±0.0 | 0.0±0.0 | 0.0±0.0 | 18.89 |
| IM[17] | 46.89±31.4 | 53.73±20.8 | 3.64±6.9 | 18.41±17.4 | 4.57±7.4 | 0.64±1.7 | 1.02±2.5 | 0.0±0.0 | 21.64 |
| CaP[20] | 60.08±17.3 | 60.11±20.24 | 8.72±9.7 | 20.33± 21.0 | 2.84±4.6 | 0.63±1.3 | 0.60±2.2 | 0.0±0.0 | 25.77 |
| **DEP** | 75.70±10.4 | 66.13±13.4 | 45.69±16.2 | 43.35±20.2 | 15.93±13.9 | 5.71±3.7 | 4.60±7.1 | 0.50±0.5 | **39.36** |
| **DEPS** | 79.77±8.5 | 79.46±10.6 | 62.40±17.9 | 53.32±29.3 | 29.24±27.3 | 13.80±8.0 | 12.56±13.3 | 0.59±0.5 | **48.56** |

In addition, *selector* also greatly improves the final task success rate of the agent (from **DEP w/o Selector** to **DEPS**). Hard meta tasks usually require the completion of multiple sub-goals (up to dozens of goals), thus bringing more flexibility and providing more candidate goals for the Selector. At the same time, as the agent conducts experiments with limited episode length, it also places high demands on the efficiency of the plan. Therefore, the Selector brings a significant improvement on efficiency-sensitive tasks such as MT7 (up to **+2.7** times success rate).

**Robustness on different controller and different Minecraft versions**  We also evaluate DEPS on MineRL [3] and MC-Textworld [22]. Note that DEPS is a planning method, which needs to equip the goal-conditioned controller for interacting with the Minecraft environments. We choose MC-Controller [6] and Steve-1 [21] as controllers to interact with Minedojo and MineRL, respectively. These two methods are all control policies that perceive visual partial observations and produce mouse and keyboard actions. While MC-Textworld is a text world, which only keeps the Minecraft crafting recipes and mining rules. So MC-Textworld does not require the controller. The DEPS results of the task set MT1-MT8 on different Minecraft environments are shown in Table 3. The results report that DEPS can generate effective plans in various Minecraft environments. The results on MC-Textworld [22] also show that the performance drops on more difficult task sets from MT6 to MT8 are mainly from the controller limitation.

Table 3: Success rates of DEPS under different Minecraft environments.

| Environment | Version | Controller | MT1 | MT2 | MT3 | MT4 | MT5 | MT6 | MT7 | MT8 |
|---|---|---|---|---|---|---|---|---|---|---|
| MineDojo [10] | 1.11.2 | [6] | 79.77 | 79.46 | 62.40 | 53.32 | 29.24 | 13.80 | 12.56 | 0.59 |
| MineRL [3] | 1.16.5 | [21] | 84.05 | 80.32 | 24.25 | 36.21 | 9.16 | 17.22 | 16.79 | 1.84 |
| MC-Textworld [22] | 1.19.2 | - | 100.00 | 90.00 | 80.00 | 56.25 | 64.71 | 57.14 | 69.57 | 50.00 |

## 4.3 Ablation Study

We conduct ablation experiments to investigate the number of candidate executable goals for different Selector models and the specific impact of the rounds of DEPS.

### 4.3.1 Ablation on Selector

We verify the robustness of our proposed Selector under different parallel goals. The agent is asked to complete 2, 3, and 4 candidate goals (the precondition is consistent for all goals), respectively. The goals of the task correspond to different kinds of mobs or materials.

We report the final success rate of our method (DEP) with different selector implementations, including using a fixed sequence of goals, a random sequence of goals, and selecting a goal based on MineCLIP [10], CLIP [35], and our horizon-predictive Selector (HPS). As Figure 4 shows, in one round of parallel candidate goals, an improvement of $\Delta$=+22.3%, +29.2%, +32.6% is obtained using our horizon-predictive Selector compared to not any selector (i.e., fixed plan), respectively.

At a limited episode length, e.g., 1000 steps, goal-model shows a greater advantage, which proves that goal-model can improve the execution efficiency of the plan in embodied environments. In addition, compared to using vision-language models such as CLIP [35] and MineCLIP [10] as a goal model, horizon-predictive has the best performance due to better estimation of the horizon information. The curve trend also demonstrates that agents with Selector scale up under large amounts of goals in an open-world environment.

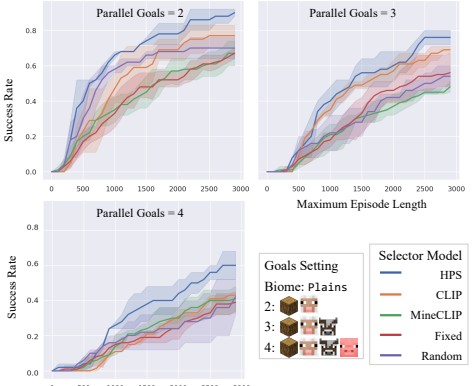

Figure 4: **The success rates of DEPS with different selectors under varying numbers of parallel goals and maximum episode lengths.**

Table 4: **Success Rate of DEPS under different maximum rounds of re-planning.** Round 0 represents the vanilla Planner w/o the re-planning process. $\infty$ represents the re-planning process will not end until task success or reaching the maximum horizon, which is still limited by the maximum tokens of LLMs. The maximum number of rounds for Codex is around 7-8 rounds.

| Rounds | 0 | 1 | 3 | 5 | $\infty$ | $\Delta$ $(0 \to \infty)$ |
|---|---|---|---|---|---|---|
| MT1 | 28.6 | 50.6 | 68.1 | 79.8 | 79.8 | **+51.2** |
| MT2 | 37.1 | 71.2 | 71.4 | 79.2 | 79.5 | **+42.4** |
| MT3 | 15.1 | 20.1 | 40.3 | 40.8 | 62.4 | **+47.3** |
| MT4 | 15.9 | 17.4 | 48.3 | 50.7 | 53.3 | **+37.4** |
| MT5 | 3.2 | 3.2 | 3.2 | 15.2 | 29.2 | **+26.0** |
| MT6 | 0.5 | 0.5 | 1.1 | 1.9 | 13.8 | **+13.3** |
| MT7 | 0.6 | 2.3 | 2.9 | 2.9 | 12.6 | **+12.0** |
| MT8 | 0.0 | 0.0 | 0.0 | 0.0 | 0.6 | **+0.6** |

#### 4.3.2 Ablation on Re-Planning Rounds

We evaluate our agent on all tasks with increasing maximum rounds of DEPS. The round is defined as a cycle of interactive LLM-based planning with description, explanation, and planning and selecting, i.e., an updated plan. All tasks for every maximum round are executed 30 times and the average success rate is reported in Table 4. We take the vanilla LLM planner as the baseline, in which the model takes the initially generated plan as the final execution plan, without involving any description, re-planning, or self-explanation processes during the task execution. Our results in the previous subsection utilize the maximum rounds possible under maximum tokens capped by OpenAI. We also report the success rate increment from vanilla planner to DEPS of every meta task in column $\Delta$ in Table 4. This set of experiments demonstrates that DEPS can iteratively improve its plan in open-world environments. More description, self-explanation, and re-planning rounds produce better results, especially for hard tasks.

### 4.4 `ObtainDiamond` Challenge

Mining diamonds in the open-world game Minecraft, i.e. MT8 in Table 2, has been a long-standing challenge for the community [14]. It is challenging because mining diamonds from scratch in Minecraft involves acquiring a sequence of difficult-to-obtain items that require complex planning on goals like mining, inventory management, crafting with and without a crafting table, tool use, smelting iron ingot in a furnace, and mining at the lowest depths. We take the `ObtainDiamond` task as a bonus experiment to show the capabilities of our zero-shot planner on complex tasks in embodied environments. Previous methods' success rates on this challenge further vouch for its difficulty. [43, 34] leverages domain-secific reward functions and RL fine-tuning to achieve $\backsim 0.1\%$ success rate in 15 minutes of game play. VPT further boosts the success rate to $20\%$ within 20 minutes of play through pre-training on collects $\backsim 70k$ hours human demonstrations and finetuning with human-designed reward function [3]. DreamerV3 is trained from scratch to collect diamonds in a modified Minecraft environment (easier to break blocks) with world models to achieve a success rate of $2\%$ [15].

Our DEPS manages to achieve on-par performance in this grand challenge; our agent achieves a 0.59% success rate within 10 minutes of gameplay. Note our method does not specifically fine-tune for this challenge. It is designed to be multi-task in its nature. Furthermore, considering our planner operates with demonstration prompts on a fixed Large Language Model, it can be straightforwardly adapted to other open-ended environments with modifications.

## 5    Related Works

**Task planning with LLMs**   There have been some methods leveraging the large language model to generate action plans for high-level tasks in embodied environments [46, 9, 11]. [16] decompose natural language commands into sequences of executable actions by text completion and semantic

translation, while SayCan generates feasible plans for robots by jointly decoding an LLM weighted by skill affordances from value functions [4]. For better executing the plan in embodied environments, some methods use an object detector describing the initial environment into the language prompt to produce environment-suitable plans and adopt success detectors to check that each step is executed successfully [17, 20]. [42] and [20] use the pythonic-style prompt to produce more executable plans. However, all of the above methods assume that the initial plan from the LLM is correct. When there are bugs in the initial plan, it's difficult for the agent to finish the task successfully.

**Interactive Planning with LLMs**    Inner Monologue [17] pilots the front of interactive planning with LLMs, which introduces the feedback (including success detection and scene description) to the planner. However, we found it could still suffer from accumulative planning error, especially in long-horizon open-world tasks. Rather, our "*Describe, Explain, Plan and Select*" (DEPS) method can produce more reliable plans by leveraging chain-of-thought thinking and explanation to locate the errors in previous plans. Moreover, we also propose a goal Selector to further improve the efficiency of the plan, thereby yielding much better performances. Readers are encouraged to refer to the comparative results in Section 4.2 between DEPS and these prior arts. There are also some concurrent works on planning with LLMs [39, 26, 23, 33, 47].

**Agents in Minecraft**    Some previous works have employed the hierarchical architecture to solve long-horizon tasks in Minecraft [30, 27, 24]. Recently, based on the internet-scale corpus, [10] pre-trains a language-conditioned reward function and learns multi-task MineAgent. [3] collects a vast amount of human demonstrations to train a behavior cloning agent. More recently, [15] utilized a learned world model to distill a policy that can efficiently explore in Minecraft. There are also some works focus on learning goal-conditioned policies for better instruction-following [6, 7, 21]. While these efforts all focus on improving the low-level controller. Rather, the planner in our architecture emphasizes applying domain knowledge to propose and arrange the sub-goals. It significantly influences the complexity and breadth of tasks that the agent can handle. Moreover, our planner is zero-shot, making it possible to generalize to other long-horizon open worlds.

# 6    Limitations

Albeit the impressive results of our approach, we believe there are at least two major limitations within our approach. First of all, our framework relies on privately-held LLMs like GPT-3 and ChatGPT, which makes it less accessible to those who cannot afford or access the service. However, we're fully committed to ensuring a more democratized method and will explore using open-sourced models including OPT [48] and BLOOM [38]. Another issue is the explicit step-by-step planning in our system. Although it brings us superior performances over the baselines, the planning bottleneck can also prevent our model from being further scaled up. A more appealing approach will be amortizing the planning within an end-to-end trainable goal-conditioned policy, which is worth exploring next. Furthermore, some previous fundamental challenges in planning (e.g., dead ends) may not prevalent in our adopted environments and hence could be inadvertently overlooked by our paper. We are dedicated to addressing more fundamental challenges present in building a multi-task generalist agent in our series of following work.

# 7    Conclusion

We investigate the problem of planning in open worlds. We identify two major challenges unique to these environments: 1) long-term planning requires precise and multi-step reasoning, and 2) planning efficiency could be compromised since canonical planners do not take the agent's proximity to parallel goals/subtasks into consideration. We propose "Describe, Explain, Plan and Select" (**DEPS**), an interactive approach based on Large Language Models (LLMs) to tackle them both. Our experiments in the challenging Minecraft domain verify the advantages of our approach over counterparts by marking the milestone of robustly accomplishing 70+ Minecraft tasks and nearly doubling the overall performances. DEPS also is the first planning-based agent that can reach the diamond in this game.

## Acknowledgements

This work is funded in part by the National Key R&D Program of China #2022ZD0160301, a grant from CCF-Tencent Rhino-Bird Open Research Fund, NSF grants #IIS-1943641, #IIS-1956441, #CCF-1837129, an SRA from Meta and a research gift from Amazon Alexa AI, and a gift from RelationalAI. We thank Dai Zhixiang from NVIDIA and Xu Hongming from BIGAI on training LLMs and infrastructure supports, respectively.

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

# Appendix

## Contents

# A  Additional Experiments

Additional experiments are conducted on the ALFWorld [41] and Tabletop Manipulation environments [40] to showcase the generalization capabilities of DEPS.

## A.1  ALFWorld

ALFWorld [41] is an interactive learning environment that aligns text and embodiment, allowing agents to acquire abstract, text-based policies in TextWorld, and subsequently execute goals from the ALFRED benchmark in a visually rich environment.

### A.1.1  Tasks

The ALFWorld framework contains six types (namely `Pick & Place`, `Examine in Light`, `Clean & Place`, `Heat & Place`, `Cool & Place`, `Pick Two & Place`) of tasks with various difficulty levels. Tasks involve first finding a particular object, which often requires the agent to open and search receptacles like drawers or cabinets. Subsequently, all tasks other than Pick & Place require some interaction with the object such as heating (place the object in a microwave and start it) or cleaning (wash the object in a sink). To complete the task, the object must be placed in the designated location. We sample 10 tasks from ALFWorld randomly and list all the task names, types, and the number of receptacles in Table 5. We classify them into 6 groups based on their functionality. For all tasks, the maximum number of steps is set as 50.

Table 5: Task list in ALFWorld.

| Group | No. | Task | Number of Receptacles |
|---|---|---|---|
| Pick & Place | 1 | put some soapbottle on garbagecan | 13 |
| | 2 | put a tissuebox in dresser | 26 |
| | 3 | put some soapbar on drawer | 15 |
| Clean & Place | 4 | put a clean soapbar in bathtubbasin | 16 |
| | 5 | clean some tomato and put it in fridge | 35 |
| Cool & Place | 6 | put a cool tomato in countertop | 30 |
| | 7 | put a cool bread in countertop | 27 |
| Heat & Place | 8 | heat some cup and put it in cabinet | 36 |
| Pick Two & Place | 9 | find two cup and put them in cabinet | 36 |
| Examine in Light | 10 | look at mug under the desklamp | 18 |

We select the GPT as Zero-Shot Planner (GPT) [16] and Inner Monologue (IM) [17] as baseline methods. For the Inner Monologue, the planning goal is the next goal among all candidate goals. For the GPT and DEP, which produce the full plan at once, the planning goal is the full plan (a goal sequence). Then the plan will be executed step-by-step, i.e., the current goal will be given to the controller and select suitable action according to the current state. The goal termination module is also employed with the LLM. For better demonstrate the effectiveness of self-explanation in DEP, we also augment the zero-shot planner with re-planning ability (GPT+RP). All planner methods access the LLM model through OpenAI API (`text-davinci-03` model [5]). Since ALFWorld is a text world, the environment will be given a literal description and candidate language-conditioned actions for each state, so the controller under ALFWorld is also LLM-based. Chain-of-Thought [45] is also employed in the controller for better decision-making. All prompts for planner and controller in ALFWorld are listed in Section G.2.

### A.1.2  Results

Each task is executed five times, and the average results for each task group are presented in Table 6. BUTLER is the a training-based method, the results are sourced from [41]. Re-planning is a crucial capability in complex and exploratory environments. The short-horizon planning approach (IM) with

**Task**: put some soapbottle on garbagecan

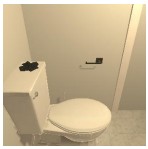 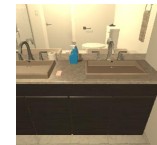 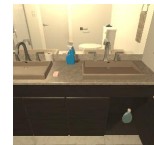 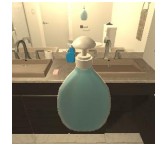 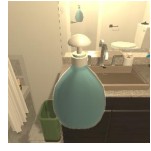 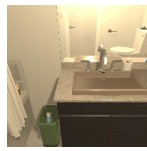

initial frame | **Goal**: pick a soapbottle at cabinet 4 **Action**: go to cabinet 4 | **Goal**: pick a soapbottle at cabinet 4 **Action**: open cabinet 4 | **Goal**: pick a soapbottle at cabinet 4 **Action**: take soapbottle 2 from cabinet 4 | **Goal**: put a soapbottle on garbagecan at garbagecan 1 **Action**: go to garbagecan 1 | **Goal**: put a soapbottle on garbagecan at garbagecan 1 **Action**: put soapbottle 2 in/on garbagecan 1

Figure 5: **Planning in the ALFWorld experiments.**

re-planning capability outperforms the long-horizon planning approach (GPT) without re-planning capability with a large margin. Furthermore, the long-horizon planning method augmented with re-planning capability (GPT+RP) achieves superior performance ranging from 10% (GPT) to 52%. DEP further enhances the feasibility of planning with descriptions and self-explanation. Notably, all planning methods fail on `Place Two & Place` tasks, which is attributable to LLM's lack of requisite knowledge for this task. It is worth investigating how to effectively incorporate the distinctive knowledge of an environment into LLM.

Table 6: Success rates of tasks in ALFWorld.

| Group | BUTLER [41] | GPT [16] | GPT+RP | IM [17] | DEP |
|---|---|---|---|---|---|
| Pick & Place | 46.0% | 33.3% | 100.0% | 33.3% | 93.3% |
| Clean & Place | 39.0% | 0.0% | 10.0% | 50.0% | 50.0% |
| Cool & Place | 100.0% | 0.0% | 30.0% | 50.0% | 100.0% |
| Heat & Place | 74.0% | 0.0% | 40.0% | 0.0% | 80.0% |
| Pick Two & Place | 24.0% | 0.0% | 0.0% | 0.0% | 0.0% |
| Examine in Light | 22.0% | 0.0% | 100.0% | 0.0% | 100.0% |
| Average | 37.0% | 10.0% | 52.0% | 30.0% | 76.0% |
| Average | .0% | 10.0% | 52.0% | 30.0% | 76.0% |

## A.2 Tabletop Manipulation

The Tabletop Manipulation experiments are conducted on a Universal Robot UR5e with a suction gripper in the simulated environments [40].

### A.2.1 Tasks

The assessment of all methods is conducted in five seen tasks, as illustrated in Table 7, wherein the seen tasks are employed for training the CLIPort [40] as the controller. The task involves a robotic arm equipped with a gripper, which is tasked with rearranging a number of blocks and bowls on a table to achieve a desired configuration specified via natural language (e.g., "putting the blocks in the bowls with matching colors").

Table 7: Task list in CLIPort.

| No | Task Name | Instruction |
|---|---|---|
| 1 | Assembling Kits | Put the objects in the corresponding holes. |
| 2 | Towers of Hanoi | Move the rings to the darker brown side. |
| 3 | Put Block in Bowl | Match the blocks and the bowls. |
| 4 | Packing Shapes | Pack the objects in the brown box. |
| 5 | Stack Block Pyramid | Stack the blocks into a pyramid. |

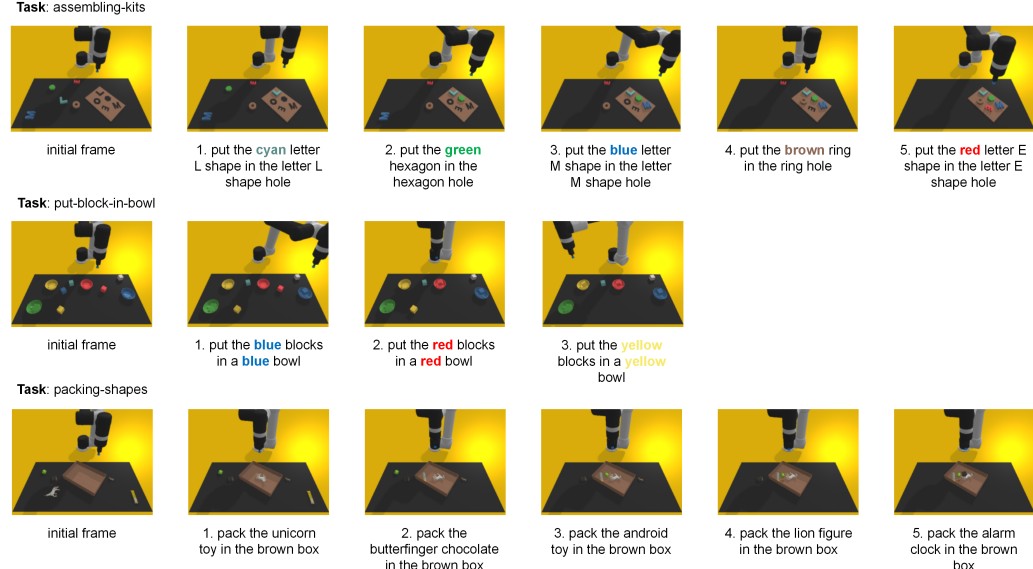

Figure 6: **Planning in the Tabletop Manipulation experiments.**

We utilized Inner Monologue (IM) [17] and Zero-shot Planner (GPT) [16] as planning baselines, in addition to comparing with a multi-task CLIPort policy directly trained on long-horizon task instructions (i.e., without utilizing LLM for planning). As CLIPort is a single-step policy that does not spontaneously terminate during policy rollout, we report CLIPort evaluations with Oracle termination (i.e., repeat until the Oracle indicates task completion) and fixed-step termination (i.e., repeat for $k$ steps). For Inner Monologue, which directly produces the next-step goal and terminates when the LLM ceases to generate new steps, we similarly set the maximum number of steps to be $k$ for practical considerations. For the zero-shot planner [16] and our DEP, which produce the full plan at once, they are augmented with the LLM-based termination. DEP also involves the description, explanation, and re-planning process. The same $k$ step is suitable for these two methods. In practice, $k$ is set as 15. The prompts for all methods are listed in Section G.3. We use the checkpoints provided by CLIPort as the controller and all planner methods access the ChatGPT (as LLM) through OpenAI API (`gpt-3.5-turbo` model). Each task is evaluated 5 times with different seeds.

### A.2.2 Results

The results of each method are listed in Table 8. All LLM-based planning methods perform well on tabletop rearrangement tasks. Given the compact nature of the tabletop environment, the performance gap among the various LLM-planning methods is not as pronounced as in the open-ended Minecraft. This observation underscores the robust generalization capabilities of LLM-based planning methods across diverse environments.

Table 8: Success rates for various methods across different tasks in Tabletop Manipulation environment.

| Task | CLIPort [40] +oracle | GPT [16] | IM [17] | DEP |
|---|---|---|---|---|
| Assembling Kits | 60.0% | 60.0% | 60.0% | 60.0% |
| Towers of Hanoi | 100.0% | 100.0% | 40.0% | 100.0% |
| Put Block in Bowl | 100.0% | 100.0% | 82.0% | 100.0% |
| Packing Shapes | 40.0% | 40.0% | 60.0% | 40.0% |
| Stack Block Pyramid | 80.0% | 100.0% | 40.0% | 100.0% |
| Average | 76.0% | 80.0% | 56.4% | 80.0% |

# B  Minecraft Task Details

To fully validate the multitask planning and execution capability of our agent, we choose over 70 tasks from the Minecraft Universe Benchmark [22] as the set of evaluation tasks. These tasks are related to items that can be obtained in the Minecraft overworld. These tasks are also a subset of MineDojo [10] programmatic tasks. Minedojo exists some programmatic tasks sharing the same object item given different conditions (e.g., obtain wool given shear or obtain wool given nothing). Minedojo expands the richness of the same tasks (sharing the same Minecraft item as an object) by giving different initial conditions (e.g., `obtain wool given shears` or `obtain wool given nothing`). We keep only the 71 hardest conditions (i.e. `given nothing`) as tasks.

We list all task names, objects, and their required skills number for planning from Table 9 to Table 16. Object item is used as the basis for the successful completion of the task. These objects cannot be obtained directly from the environment, and usually require multiple goals (i.e., reasoning steps) to be constructed. Here we only consider the number of required goal types, and multiple identical goals are unified into 1 reasoning step. Note that the reasoning steps for each task are not fixed, and as the initial state of the agent and the biome is in change, more reasoning steps may be required to complete it, we only report the most basic case here.

As shown in Figure 4, for each task, a relaxed (longer) maximum episode steps will increase the success rate of the task. To fully test the efficiency of our method, we set an upper limit on the episode length for each task. Since different tasks have different difficulty levels, we double the average completion time of human players for different meta-tasks as the upper limit of the episode. The play time are computed as corresponding maximum steps (i.e., Max. Steps in Table 1) of episode length at 20Hz.

Table 9: Task details on MT1 **Basic** set.

| Meta-Task | ID | Task Name | Required Skills | Object | Initial Inventory | Instruction |
|---|---|---|---|---|---|---|
| MT1 Basic | 1 | CraftPlanks | 2 | planks | null | Obtain a plank. |
| | 2 | CraftSticks | 3 | stick | | Obtain a stick. |
| | 3 | CraftWoodenSlab | 4 | wooden_slab | | Obtain a wooden slab. |
| | 4 | CraftWoodenPressure | 3 | wooden_pressure | | Obtain a wooden pressure plate. |
| | 5 | CraftBowl | 4 | bowl | | Obtain a bowl. |
| | 6 | CraftWoodenButton | 3 | wooden_button | | Obtain a wooden button. |
| | 7 | CraftChest | 4 | chest | | Obtain a chest. |
| | 8 | CraftOakStairs | 4 | oak_stairs | | Obtain an oak stair. |
| | 9 | CraftSign | 5 | sign | | Obtain a sign. |
| | 10 | CraftFence | 5 | fence | | Obtain a fence. |
| | 11 | CraftFenceGate | 5 | fence_gate | | Obtain a fence gate. |
| | 12 | CraftBoat | 4 | boat | | Obtain a boat. |
| | 13 | CraftTrapdoor | 4 | trapdoor | | Obtain a trap door. |
| | 14 | CraftWoodenDoor | 4 | door | | Obtain a door. |

Table 10: Task details on MT2 **Tool (Simple)** set.

| Meta-Task | ID | Task Name | Required Skills | Object | Initial Inventory | Instruction |
|---|---|---|---|---|---|---|
| MT2 Tool (Simple) | 15 | CraftCraftingTable | 3 | crafting_table | null | Obtain a crafting table. |
| | 16 | CraftWoodenPickaxe | 5 | wooden_pickaxe | | Obtain a wooden pickaxe. |
| | 17 | CraftWoodenAxe | 5 | wooden_axe | | Obtain a wooden axe. |
| | 18 | CraftWoodenHoe | 5 | wooden_hoe | | Obtain a wooden hoe. |
| | 19 | CraftWoodenSword | 5 | wooden_sword | | Obtain a wooden sword. |
| | 20 | CraftWoodenShovel | 5 | wooden_shovel | | Obtain a wooden shovel. |
| | 21 | CraftFurnace | 7 | furnace | | Obtain a furnace. |
| | 22 | CraftStonePickaxe | 7 | stone_pickaxe | | Obtain a stone pickaxe. |
| | 23 | CraftStoneAxe | 7 | stone_axe | | Obtain a stone axe. |
| | 24 | CraftStoneHoe | 7 | stone_hoe | | Obtain a stone hoe. |
| | 25 | CraftStoneShovel | 7 | stone_shovel | | Obtain a stone shovel. |
| | 26 | CraftStoneSword | 7 | stone_sword | | Obtain a stone sword. |

Table 11: Task details on MT3 **Hunt and Food** set.

| Meta-Task | ID | Task Name | Required Skills | Object | Initial Inventory | Instruction |
|---|---|---|---|---|---|---|
| MT3 Hunt & Food | 27 | CraftBed | 5 | bed | | Obtain a bed. |
| | 28 | CraftPainting | 6 | painting | | Obtain a painting. |
| | 29 | CraftCarpet | 5 | carpet | | Obtain a carpet. |
| | 30 | CraftItemFrame | 6 | item_frame | null | Obtain an item frame. |
| | 31 | CookPorkchop | 9 | cooked_porkchop | | Cook the porkchop. |
| | 32 | CookBeef | 9 | cooked_beef | | Cook the beef. |
| | 33 | CookMutton | 9 | cooked_mutton | | Cook the mutton. |

Table 12: Task details on MT4 **Dig-Down** set.

| Meta-Task | ID | Task Name | Required Skills | Object | Initial Inventory | Instruction |
|---|---|---|---|---|---|---|
| MT4 Dig-Down | 34 | CraftStoneStairs | 7 | stone_stairs | | Obtain a stone stair. |
| | 35 | CraftStoneSlab | 7 | stone_slab | | Obtain a stone slab. |
| | 36 | CraftArmorStand | 10 | armor_stand | | Obtain an armor stand. |
| | 37 | CraftCobblestoneWall | 7 | cobblestone_wall | | Obtain a cobblestone wall. |
| | 38 | CraftQuartzBlock | 10 | quartz_block | | Obtain a quartz block. |
| | 39 | CraftStoneBrick | 9 | stone_brick | | Obtain a stone brick. |
| | 40 | SmeltStone | 9 | stone | null | Smelt a stone. |
| | 41 | CraftTorch | 9 | torch | | Obtain a stone brick. |
| | 42 | ObtainCoal | 8 | coal | | Mine coal. |
| | 43 | CraftStoneBrickStairs | 10 | stonebrick_stairs | | Obtain a stone brick. |
| | 44 | CraftStonePressurePlate | 9 | stone_pressure_plate | | Obtain a stone brick. |
| | 45 | CraftStoneButton | 7 | stone_button | | Obtain a stone brick. |
| | 46 | CraftLever | 7 | level | | Obtain a stone brick. |

Table 13: Task details on MT5 **Equipment** set.

| Meta-Task | ID | Task Name | Required Skills | Object | Initial Inventory | Instruction |
|---|---|---|---|---|---|---|
| MT5 Equipment | 47 | EquipLeatherBoots | 5 | leather_boots | | Equip the leather boot. |
| | 48 | EquipLeatherChestplate | 5 | leather_chestplate | | Equip the leather chestplate. |
| | 49 | EquipLeatherHelmet | 5 | leather_helmet | | Equip the leather helmet. |
| | 50 | EquipLeatherLeggings | 5 | leather_leggings | | Equip the leather leggings. |
| | 51 | EquipShield | 11 | shield | null | Equip the shield. |
| | 52 | EquipIronChestplate | 11 | iron_chestplate | | Equip the iron chestplate. |
| | 53 | EquipIronLeggings | 11 | iron_leggings | | Equip the iron leggings. |
| | 54 | EquipIronHelmet | 11 | iron_helmet | | Equip the iron helmet. |
| | 55 | EquipIronBoots | 11 | iron_boots | | Equip the iron boots. |

Table 14: Task details on MT6 **Tool (Complex)** set.

| Meta-Task | ID | Task Name | Required Skills | Object | Initial Inventory | Instruction |
|---|---|---|---|---|---|---|
| MT6 Tool (Complex) | 56 | CraftBucket | 11 | bucket | | Obtain a bucket. |
| | 57 | CraftShears | 11 | shears | | Make shears. |
| | 58 | CraftIronPickaxe | 11 | iron_pickaxe | | Obtain an iron pickaxe. |
| | 59 | CraftIronAxe | 11 | iron_axe | null | Obtain an iron axe. |
| | 60 | CraftIronHoe | 11 | iron_hoe | | Obtain an iron hoe. |
| | 61 | CraftIronShovel | 11 | iron_shovel | | Obtain an iron shovel. |
| | 62 | CraftIronSword | 11 | iron_sword | | Obtain an iron sword. |

Table 15: Task details on MT7 **Iron-Stage** set.

| Meta-Task | ID | Task Name | Required Skills | Object | Initial Inventory | Instruction |
|---|---|---|---|---|---|---|
| | 63 | CraftIronBars | 11 | iron_bars | | Obtain an iron bar. |
| | 64 | CraftIronNugget | 11 | iron_nugget | | Obtain an iron nugget. |
| | 65 | CraftMinecart | 11 | minecart | | Obtain a minecart. |
| | 66 | CraftHopper | 12 | hopper | | Obtain a hopper. |
| | 67 | CraftHopperMinecart | 14 | hopper_minecart | | Obtain a hopper minecart. |
| | 68 | CraftFurnaceMinecart | 12 | furnace_minecart | | Obtain a furnace minecart. |
| MT7 | 69 | CraftCauldron | 11 | cauldron | null | Obtain a cauldron. |
| Iron-Stage | 70 | CraftChestMinecart | 13 | chest_minecart | | Obtain a chest minecart. |
| | 71 | CraftIronDoor | 11 | iron_door | | Obtain an iron door. |
| | 72 | CraftIronTrapdoor | 11 | iron_trapdoor | | Obtain an iron trapdoor. |
| | 73 | CraftTripwireHook | 11 | tripwire_hook | | Obtain a tripwire hook. |
| | 74 | CraftHWPressurePlate | 11 | heavy_weighted_plate | | Obtain a heavy weighted plate. |
| | 75 | CraftRail | 11 | rail | | Obtain a rail. |

Table 16: Task details on MT8 **Challenge** set.

| Meta-Task | ID | Task Name | Required Skills | Object | Initial Inventory | Instruction |
|---|---|---|---|---|---|---|
| Challenge MT8 | 76 | ObtainDiamond | 12 | diamond | null | Obtain a diamond. |

# C  DEPS Implementation Details

We study three different implementations of DEPS for each of the experimental settings. While each version incorporates description and self-explanation to improve planning of LLM, there are differences in the internal components of each system, as seen in Table 17.

Table 17: Comparison between different versions of DEPS implemented in three different environments.

| | **Minecraft** | **ALFWorld** | **Tabletop Manipulation** |
|---|---|---|---|
| LLM | code-davinci-02 | text-davinci-03 | gpt-3.5-turbo |
| Controller | Behavior Cloning Learned | LLM-based | CLIPort |
| Descriptor | Inventory Description | Env Support | heuristics |
| Explainer | LLM-based | LLM-based | LLM-based |
| Selector | Horizon Prediction Module | N/A | N/A |

## C.1  Controller

As the name implies, tasks in Minecraft are usually related to mine and craft goals. Mine goals require the agent to collect raw materials from the environment using the appropriate tools. Craft goals ask the agent to synthesize using existing materials. Any raw material used requires the agent to collect through suitable tools (e.g., diamonds can only be collected by an iron pickaxe or a better pickaxe). So a task usually requires dozens of step-by-step mine and craft goals, as the required skills in Table 9. Note that the successful execution of a task needs to satisfy certain exact numerical constraints due to the presence of strict generation recipes in the environment (e.g., a log can craft 4 planks, so harvesting 6 planks requires at least 2 logs). When the number of materials collected is not enough, the goal cannot be completed successfully. When more materials are collected than actually needed, the execution success rate of the task could also be reduced because the plan can not be finished under the maximum action steps.

Table 18: The success rate of different skill/goal with imitation learning controller.

| ID | Skill Description | Success Rate | Episode Length |
|----|-------------------|--------------|----------------|
| 0 | Mine 1 oak wood | 0.39 | 600 |
| 1 | Mine birch wood | 0.29 | 600 |
| 2 | Mine 1 cobblestone with pickaxe | 0.95 | 600 |
| 3 | Mine 1 stone with pickaxe | 0.70 | 600 |
| 4 | Mine 1 seed | 0.18 | 600 |
| 5 | Mine 1 leaves with shears | 0.68 | 600 |
| 6 | Mine 1 dirt | 0.54 | 600 |
| 7 | Mine 1 iron ore with stone pickaxe | 0.40 | 3000 |
| 8 | Mine 3 iron ore with stone pickaxe | 0.16 | 3000 |
| 9 | Mine 1 diamond with iron pickaxe | 0.35 | 12000 |
| 10 | Mine 1 diamond with stone pickaxe | 0.00 | 12000 |
| 11 | Kill 1 sheep with axe | 0.44 | 600 |
| 12 | Kill 1 cow with axe | 0.60 | 600 |
| 13 | Kill 1 chicken with axe | 0.46 | 600 |
| 14 | Kill 1 pig with axe | 0.49 | 600 |
| 15 | Kill 1 llama | 0.50 | 600 |
| 16 | Equip tool on mainhand | 1.00 | 600 |
|  | Craft w/o crafting_table | 1.00 | 600 |
| 17-261 | Craft w/ crafting_table | 0.90 | 600 |
|  | Smelt w/ furnace | 0.80 | 600 |

We designed the agent's skill space based on these goals, as shown in Table 18, with a total of 262 goals. Every goal is designed with an objective item (e.g., 1 `minecraft:cobblestone` for skill "`Mine 1 cobblestone with pickaxe`"), which is used to evaluate the achievement of the goal. The skill, as a goal-conditioned policy $\pi(a|s, g)$ for decision-making, maps the current state $s$ and goal $g$ to action $a$. The goal is specified as natural language instructions here, which is similar to [4].

When training the controller, we adopt the observation space provided by MineDoJo [10], which includes an RGB camera view, yaw/pitch angle, GPS location, and the type of $3 \times 3$ blocks surrounding the agent. We discretize the original multi-discrete action space provided by MineDojo into 42 discrete actions. We use the proposed imitation learning method proposed by [6] in training. To be specific, a modified goal-sensitive Impala CNN is used as the backbone network. The success rate under a fixed episode length of every skill is listed in Table 18.

## C.2   LLM as Planner

DEPS relies on Large Language Models (LLMs) to generate language-based plans. In our Minecraft experiment, we chose Codex [8] as the LLM Planner because it can accept longer input tokens and is cost-effective. However, DEPS is compatible with various types of LLMs. Therefore, we used GPT3 [5] and ChatGPT as LLM Planners in the ALFWorld and Tabletop Manipulation experiments, respectively. Due to the effective planning and error correction performance of DEPS, the initial plan generated by the LLM has little impact on the final performance of the Agent. We also conduct ablation experiments on

even if the initial plan generated by the LLM has low accuracy, DEPS can generate a final feasible plan through self-explanation and re-planning. Therefore, we conducted ablation experiments on LLM in Minecraft.

We choose Codex [8], ChatGPT, GPT3 [5], and recent GPT-4 [31] as Planners. We used Vanilla Planner [16] as baselines and excluded the re-planning process. Given the same prompt with DEPS, the performance of baseline models reflects the planning ability of different LLMs. The success rate of baseline and DEPS on different LLMs are reported in Table 19.

Table 19: Success rates for different LLMs on Minecraft tasks.

| Group | Codex [8] | | GPT-3 [5] | | ChatGPT | | GPT-4 [31] | |
|---|---|---|---|---|---|---|---|---|
| | baseline | DEPS | baseline | DEPS | baseline | DEPS | baseline | DEPS |
| MT1 | 28.6 | 79.8 | 27.2 | 75.4 | 20.3 | 70.2 | 49.2 | 89.3 |
| MT2 | 37.1 | 79.5 | 42.1 | 76.3 | 28.2 | 68.5 | 48.3 | 85.0 |
| MT3 | 15.1 | 62.4 | 7.8 | 58.7 | 3.2 | 50.4 | 38.04 | 63.4 |
| MT4 | 15.9 | 53.3 | 6.7 | 50.2 | 4.8 | 47.8 | 27.0 | 55.7 |
| MT5 | 3.2 | 29.2 | 2.7 | 17.2 | 0.8 | 16.3 | 15.7 | 32.2 |
| MT6 | 0.5 | 13.8 | 0.3 | 7.9 | 0.3 | 6.0 | 4.9 | 16.19 |
| MT7 | 0.6 | 12.6 | 0.4 | 5.3 | 0.5 | 5.2 | 3.1 | 16.41 |

The success rate of Vanilla Planner varies on the LLMs. The GPT-4 baseline achieved an initial plan accuracy twice as high as the baselines on other LLMs, demonstrating superior planning ability. After being augmented by Descriptor, Explainer, and Selector, DEPS based on different LLMs showed almost identical success rates. This indicates that DEPS-augmented LLMs can generate more feasible plans in open-world environments even if the initial plan is less successful.

It is noteworthy that DEPS is constrained by the maximum token limits of various models, which dictate the maximum re-planning rounds that can be supported. Longer re-planning rounds tend to yield superior performance, particularly in long-horizon tasks requiring more skills (in MT6-MT7), as detailed in the Section 4.3.

Since we use pretrained LLM as a planner, it indeed requires exposure to a large amount of Minecraft-related corpus during the pretraining phase. Considering that Minecraft is one of the most popular games worldwide, there is relatively abundant data about Minecraft available online. We conducted experiments using open-source pretrained LLaMA2-70B on several Minecraft tasks and found that DEPS based on LLaMA2 also performs reliable planning under Minecraft conditions. Considering limited training data used by LLaMA2, we further finetuned an open-source language model (LLaMA2-13B) using Minecraft texts obtained from the internet which exhibited better planning performance. The results are shown in Table 20.

Table 20: Results of DEPS based on open-sourced LLaMA language models.

| Language Model | CraftingTable | WoodenPickaxe | Furnace | StonePickaxe |
|---|---|---|---|---|
| Pretrained LLaMA2-70B [44] | 60.0 | 50.0 | 40.0 | 50.0 |
| Finetuned LLaMA2-13B [44] | 90.0 | 80.0 | 70.0 | 80.0 |
| OpenAI Codex [8] | 90.0 | 80.0 | 66.7 | 73.3 |

### C.3  LLM as Explainer

Given the description and previous plan, the explainer can generate a self-explanation of the failure of the current plan and give instructions to fix the bugs. The explainer is implemented with the OpenAI completion mode based on `text-davinci-03` models. The prompt for the explainer is listed in Listing 1.

```
Here are some actions that the agent fails to perform in Minecraft. Please give
    the explanation of action execution failure according to the current inventory
     information of the agent.

###
Failed Action: mine({'iron_ore':1}, null); # step 5: mine 1 iron_ore without tool
Current Inventory: null
Explanation: Because mining iron_ore needs to use the tool stone_pickaxe, but my
    inventory does not have stone_pickaxe. So I need to craft stone_pickaxe first.

###
Failed Action: craft({'stone_pickaxe':1}, {'cobblestone':3, 'stick':2}, '
    crafting_table'); # step 1: craft 1 stone_pickaxe from 3 cobblestone and 2
    stick, on crafting_table
Current Inventory: null
Explanation: Because crafting stone_pickaxe needs to have 3 cobblestone and 2
    stick in inventory, but my inventory does not have cobblestone and stick. So I
     need to mine cobblestone and craft stick first.
```

```
###
Failed Action: craft({'stick':4}, {'planks':2}, null); # step 3: craft 4 stick
    from 2 planks first
Current Inventory: null
Explanation: Because crafting stick needs to have planks in inventory, but my
    inventory does not have planks. So I need to craft planks first.

###
```

Listing 1: Prompt for Explainer in Minecraft tasks

### C.4 Other modules

**Goal Parser** We need to map the plan expressed in free-form language to the pre-defined controller skills set. We use the LLM as an automatic parser to parse the language plan first. For the goals not following pre-defined code expression, we calculate its semantic distance to the skills by cosine similarity with pre-trained Sentence-Bert model [37] and select the most similar skill as the corresponding goal. All executable goals are listed in Appendix C. The LLM-based parser is general and can be transferred to other domains easily by modifying the prompt. The prompt for Minecraft parser is listed in Listing 2.

```
Extract the action name, action type, goal object, tool and action rank from the
    input text.

input: mine({'log':3}, null); # step 1: mine 3 log without tool
name: mine_log
action: mine
object: {'log':3}
tool: null
rank: 1
###

input: craft({'planks':12}, {'log':3}, null); # step 2: craft 12 planks from 3 log
name: craft_planks
action: craft
object: {'planks':12}
materials: {'log':3}
tool: null
rank: 2
###
```

Listing 2: Prompt for Goal Parser in Minecraft tasks

**Success Detector** The successful execution of a plan is contingent upon the agent's perception of the current goal's completion status, which is assessed by the success detector. In Minecraft, agents possess an inventory that contains all pertinent information regarding the agent's current state. Thus, the Success Detector can be implemented by monitoring changes in object information within the item inventory. In other scenarios, we can query the LLM to ascertain whether the agent has accomplished a general goal by describing the agent's current state. Alternatively, in certain environments [40], the execution of a goal is linked to the agent's current reward, signifying that these rewards can serve as automatic success detectors.

**Prompt** The generalization of the LLM to different tasks relies on well-designed prompts and related demonstrations [28]. Given an instruction command (e.g., `ObtainDiamond`) as task $T$, a prompt generator (ProG) will translate $T$ into prompt text. We also added two DEP examples in the prompt as demonstrations to make the LLM output familiar to the chain-of-thought thinking and structural output. We also design a chain-of-thought code-comments-type planning prompt to better demonstrate the capabilities of LLM. All messages are modified to suitable prompts through the prompt-generator before being input to LLM, including task $T$ and description $d_t$. The full prompt sentences and interaction logs are listed in Appendix H.

## D Comparison with other LLM-based Planners

The architectures of the different LLM-based planners are illustrated in Figure 7. Where (b) describes the information in the environment into LLM via scene descriptor and success detector, and directly

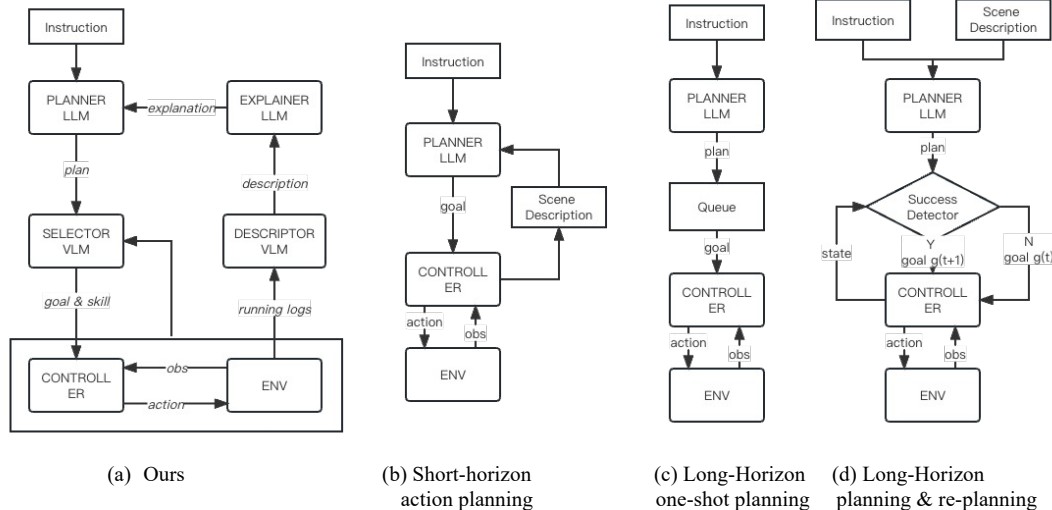

| (a) Ours | (b) Short-horizon action planning | (c) Long-Horizon one-shot planning | (d) Long-Horizon planning & re-planning |

Figure 7: **Comparison of LLM-based planner architecture**. (a), (b), (c), (d) represents planner of ours, Inner Monologue [17], Zero-Shot Planner [16] and Zero-Shot Planner with re-planning process, respectively.

plans the next goal/action, (c) is Zero-Shot planner [16], which generates the step-by-step goal sequences as plan and ignores the environment state and execution feedback, (d) is the Zero-Shot planner augmented with textual feedback and re-planning process. DEPS further rethink and explain the feedback of previous plans explicitly with the descriptor and explainer. The LLM-based planner will re-plan the task according to the explanation, as demonstrated in Figure 7(a). In addition, the goal Selector further improves the executability of the LLM plan.

# E Discussion on `ObtainDiamond` Task

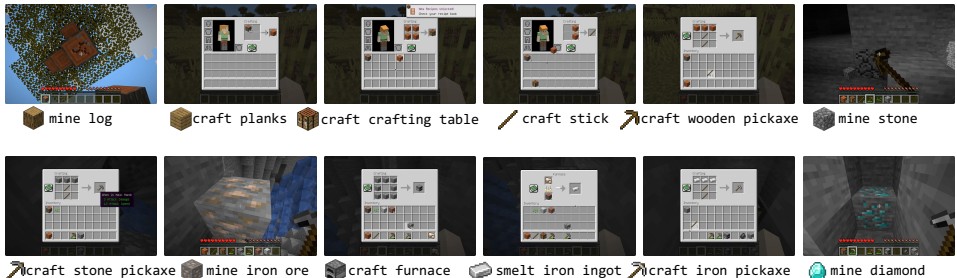

Figure 8: **The milestone goals of the `ObtainDiamond` task.**

As outlined in Section 4.4, `ObtainDiamond` task is a formidable task within the open-ended Minecraft environment. Given the necessity to explore an infinitely expansive world, an efficient plan can prove advantageous, as shown in Figure 8. The task is allotted a maximum of 12,000 steps to interact with the environment, which is comparable to that of human performance [14]. Rather than manually devising explicit hierarchical rewards, we opt to utilize DEPS for generating a hierarchical plan, which is then transferred to the downstream controller to progressively achieve each goal. When equipped with an **Oracle** Controller, DEPS yields a success rate of 60% for ObtainDiamond. In our experimentation, we employed Behavior Cloning to train a Controller agent [6]. DEPS+BC Controller achieved a success rate of 0.6% in randomly generated Minecraft worlds. The primary bottleneck impeding overall agent success rate lies within the goal-conditioned Controller, not the plans generated by DEPS. Thus, it is worth exploring the development of a data-efficient Controller capable of accepting Language goals.

Another rationale for using DEPS is that, akin to reality, materials in Minecraft possess quantity constraints, and durability for tools. In `ObtainDiamond` task, an iron pickaxe is typically insufficient

to support the agent, given the rarity of diamonds within the environment (which are predominantly found between depths of 2-16 layers and appear only 0.0846% of the time). The robust re-planning capabilities of DEPS can facilitate the generation of a feasible plan (initiating with crafting an iron-pickaxe) based on the agent's current state.

Additionally, we report the milestones, which demonstrate the decreasing success rate of subsequent tasks in Figure 9 attributable to the task's inherent complexity and Controller constraints.

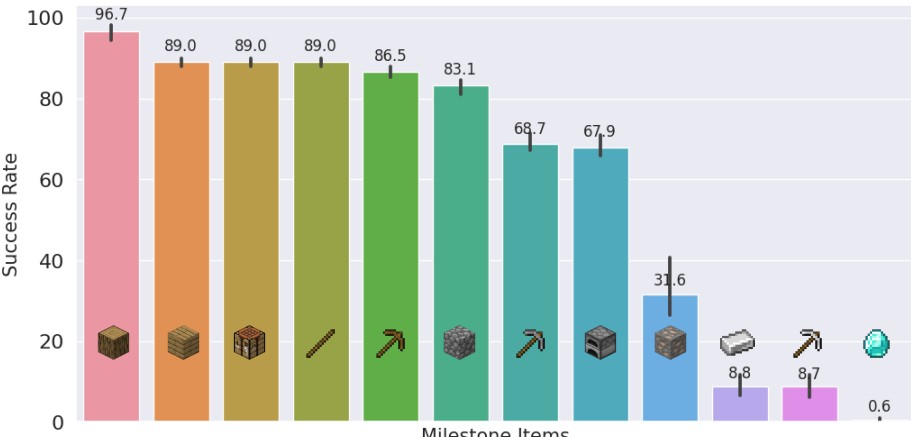

Figure 9: **Success rate of milestone items for mining diamond**.

# F    Success Rates of ALL Tasks in Minecraft

We report the complete and detailed success rate table of all tasks for different methods in Table 21, including Zero-shot Planner [16], ProgPrompt [42], Chain-of-Thought [45], Inner Monologue [17], Code as Policies [20], and proposed methods (i.e., DEP w/o Selector, and DEPS).

All tasks are executed for at least 30 times across different world seeds, given the same initial conditions. The birth positions of the world are random according to the seed. The average success rates are listed in Table 21. Our approach is state-of-the-art on almost all tasks, especially on difficult tasks that require more skills.

Table 21: Success rate comparison of various methods on MineDojo [10] environments.

| Meta-Task | Task Object | GPT [16] | PP [42] | CoT [45] | IM [17] | CaP [20] | DEP | DEPS |
|---|---|---|---|---|---|---|---|---|
| Basic MT1 | planks | 56.7 | 56.7 | 83.3 | 83.3 | 83.3 | 83.3 | 83.3 |
| | stick | 0.0 | 56.7 | 83.3 | 83.3 | 83.3 | 83.3 | 86.7 |
| | wooden_slab | 26.7 | 26.7 | 56.7 | 83.3 | 83.3 | 83.3 | 76.7 |
| | wooden_button | 23.3 | 50.0 | 73.3 | 73.3 | 73.3 | 73.3 | 96.7 |
| | wooden_pressure_plate | 80.0 | 80.0 | 53.3 | 80.0 | 80.0 | 80.0 | 86.7 |
| | chest | 0.0 | 26.7 | 0.0 | 0.0 | 0.0 | 50.0 | 76.7 | 76.7 |
| | oak_stairs | 20.0 | 40.0 | 36.7 | 16.7 | 36.7 | 56.7 | 60.0 |
| | sign | 23.3 | 0.0 | 43.3 | 0.0 | 43.3 | 63.3 | 86.7 |
| | fence | 20.0 | 20.0 | 0.0 | 20.0 | 43.3 | 63.3 | 80.0 |
| | fence_gate | 63.3 | 0.0 | 63.3 | 63.3 | 63.3 | 93.3 | 73.3 |
| | boat | 0.0 | 0.0 | 0.0 | 26.7 | 56.7 | 83.3 | 73.3 |
| | trapdoor | 26.7 | 26.7 | 26.7 | 56.7 | 56.7 | 83.3 | 76.7 |
| | bowl | 0.0 | 23.3 | 0.0 | 23.3 | 46.7 | 70.0 | 80.0 |
| | wooden_door | 23.3 | 23.3 | 46.7 | 46.7 | 46.7 | 66.7 | 80.0 |
| Tool(Simple) MT2 | crafting_table | 70.0 | 23.3 | 70.0 | 70.0 | 70.0 | 70.0 | 90.0 |
| | wooden_pickaxe | 80.0 | 80.0 | 80.0 | 80.0 | 80.0 | 80.0 | 80.0 |
| | wooden_axe | 46.7 | 46.7 | 70.0 | 70.0 | 70.0 | 70.0 | 96.7 |
| | wooden_hoe | 86.7 | 56.7 | 86.7 | 30.0 | 86.7 | 86.7 | 86.7 |
| | wooden_sword | 83.3 | 83.3 | 83.3 | 83.3 | 83.3 | 83.3 | 86.7 |
| | wooden_shovel | 76.7 | 76.7 | 76.7 | 76.7 | 76.7 | 76.7 | 90.0 |
| | furnace | 20.0 | 20.0 | 0.0 | 40.0 | 40.0 | 60.0 | 66.7 |
| | stone_pickaxe | 16.7 | 16.7 | 36.7 | 36.7 | 53.3 | 53.3 | 73.3 |
| | stone_axe | 0.0 | 0.0 | 30.0 | 30.0 | 30.0 | 46.7 | 70.0 |
| | stone_hoe | 20.0 | 20.0 | 36.7 | 36.7 | 56.7 | 56.7 | 66.7 |
| | stone_shovel | 56.7 | 56.7 | 40.0 | 36.7 | 36.7 | 56.7 | 66.7 |
| | stone_sword | 16.7 | 0.0 | 53.3 | 53.3 | 36.7 | 53.3 | 80.0 |
| Hunt and Food MT3 | bed | 16.7 | 23.3 | 23.3 | 6.7 | 6.7 | 23.3 | 43.3 |
| | painting | 33.3 | 0.0 | 0.0 | 16.7 | 16.7 | 53.3 | 86.7 |
| | carpet | 0.0 | 33.3 | 0.0 | 0.0 | 13.3 | 33.3 | 43.3 |
| | item_frame | 23.3 | 50.0 | 23.3 | 0.0 | 23.3 | 73.3 | 83.3 |
| | cooked_porkchop | 0.0 | 0.0 | 0.0 | 0.0 | 0.0 | 40.0 | 50.0 |
| | cooked_beef | 0.0 | 0.0 | 0.0 | 0.0 | 0.0 | 53.3 | 63.3 |
| | cooked_mutton | 0.0 | 13.3 | 0.0 | 0.0 | 0.0 | 43.3 | 66.7 |
| Dig-down MT4 | stone_stairs | 16.7 | 23.3 | 20.0 | 36.7 | 16.7 | 56.7 | 66.7 |
| | stone_slab | 16.7 | 50.0 | 0.0 | 16.7 | 33.3 | 50.0 | 73.3 |
| | cobblestone_wall | 16.7 | 16.7 | 16.7 | 16.7 | 43.3 | 43.3 | 63.3 |
| | lever | 0.0 | 0.0 | 0.0 | 46.7 | 46.7 | 70.0 | 83.3 |
| | coal | 0.0 | 16.7 | 0.0 | 6.7 | 0.0 | 16.7 | 20.0 |
| | torch | 0.0 | 6.7 | 0.0 | 6.7 | 0.0 | 23.3 | 13.3 |
| Equipment MT5 | leather_boots | 0.0 | 13.3 | 0.0 | 13.3 | 13.3 | 36.7 | 60.0 |
| | leather_chestplate | 0.0 | 6.7 | 16.7 | 0.0 | 6.7 | 23.3 | 36.7 |
| | leather_helmet | 16.7 | 6.7 | 0.0 | 6.7 | 0.0 | 26.7 | 70.0 |
| | leather_leggings | 0.0 | 0.0 | 0.0 | 20.0 | 0.0 | 30.0 | 56.7 |
| | iron_chestplate | 0.0 | 0.0 | 0.0 | 0.0 | 0.0 | 0.0 | 0.0 |
| | iron_leggings | 0.0 | 0.0 | 0.0 | 0.0 | 0.0 | 3.3 | 3.3 |
| | iron_helmet | 0.0 | 0.0 | 0.0 | 0.0 | 0.0 | 0.0 | 3.3 |
| | iron_boots | 0.0 | 0.0 | 0.0 | 0.0 | 0.0 | 6.7 | 20.0 |
| | shield | 0.0 | 0.0 | 0.0 | 0.0 | 6.7 | 16.7 | 13.3 |
| Tool Complex MT6 | bucket | 0.0 | 3.3 | 0.0 | 0.0 | 3.3 | 10.0 | 6.7 |
| | shears | 0.0 | 0.0 | 0.0 | 0.0 | 0.0 | 10.0 | 30.0 |
| | iron_pickaxe | 0.0 | 0.0 | 0.0 | 6.7 | 0.0 | 6.7 | 10.0 |
| | iron_axe | 0.0 | 0.0 | 0.0 | 0.0 | 0.0 | 0.0 | 16.7 |
| | iron_hoe | 0.0 | 0.0 | 0.0 | 0.0 | 0.0 | 3.3 | 13.3 |
| | iron_shovel | 0.0 | 0.0 | 0.0 | 0.0 | 0.0 | 6.7 | 13.3 |
| | iron_sword | 0.0 | 0.0 | 0.0 | 0.0 | 3.3 | 3.3 | 6.7 |
| Iron-Stage MT7 | iron_bars | 0.0 | 0.0 | 0.0 | 0.0 | 0.0 | 0.0 | 6.7 |
| | iron_nugget | 0.0 | 6.7 | 0.0 | 6.7 | 6.7 | 23.3 | 40.0 |
| | minecart | 0.0 | 0.0 | 0.0 | 0.0 | 0.0 | 3.3 | 10.0 |
| | hopper | 0.0 | 0.0 | 0.0 | 0.0 | 0.0 | 3.3 | 6.7 |
| | hopper_minecart | 0.0 | 0.0 | 0.0 | 0.0 | 0.0 | 0.0 | 0.0 |
| | furnace_minecart | 0.0 | 0.0 | 0.0 | 0.0 | 0.0 | 3.3 | 3.3 |
| | chest_minecart | 0.0 | 0.0 | 0.0 | 0.0 | 0.0 | 3.3 | 3.3 |
| | iron_door | 0.0 | 0.0 | 0.0 | 0.0 | 0.0 | 0.0 | 3.3 |
| | iron_trapdoor | 0.0 | 0.0 | 0.0 | 0.0 | 0.0 | 3.3 | 16.7 |
| | tripwire_hook | 0.0 | 0.0 | 0.0 | 6.7 | 0.0 | 16.7 | 30.0 |
| | heavy_weighted_pressure_plate | 0.0 | 0.0 | 0.0 | 0.0 | 0.0 | 3.3 | 33.3 |
| | rail | 0.0 | 0.0 | 0.0 | 0.0 | 0.0 | 0.0 | 6.7 |
| | cauldron | 0.0 | 0.0 | 0.0 | 0.0 | 0.0 | 0.0 | 3.3 |
| MT8 Challenge | diamond | 0.0 | 0.0 | 0.0 | 0.0 | 0.0 | 0.5 | 0.6 |

# G   Prompt for Different Tasks and Different Methods

## G.1   Prompt for Minecraft Tasks

### G.1.1   DEPS

```
User: You are a helper agent in Minecraft. You need to generate the sequences of
    goals for a certain task in Minecraft.
Planner: OK.

==========
User: How to obtain 1 wooden_button in Minecraft step-by-step?
Planner: The code for obtaining 1 wooden_button is as follows:
def obtain_1_wooden_button(inventory={}):
   mine({'log':1}, null); # step 1: mine 1 log without tool
   craft({'planks':4}, {'log':1}, null); # step 2: craft 4 planks from 1 log
   craft({'wooden_button':1}, {'planks':1}, null); # step 3: craft 1 wooden_button
       from 1 planks
   return "wooden_button"
Descriptor: I succeed in step 1, 2, 3.
User: Good. I finish all steps and I obtain 1 stone_pickaxe successfully.
Planner: OK.

==========
User: How to obtain 1 stone_sword in Minecraft step-by-step?
Planner: The code for obtaining stone_sword is as follows:
def obtain_1_stone_sword(inventory = {}):
   mine({'log':3}, null); # step 1: mine 3 log without tool
   craft({'planks':12}, {'log':3}, null); # step 2: craft 12 planks from 3 log
   craft({'stick':4}, {'planks':2}, null); # step 3: craft 4 stick from 2 planks
   mine({'cobblestone':2}, null); # step 4: mine 2 cobblestone without tool
   craft({'stone_sword':1}, {'cobblestone':2, 'stick':1}, 'crafting_table'); #
       step 5: craft 1 stone_sword from 2 cobblestone and 1 stick, on
       crafting_table
   return 'stone_sword'

Descriptor: I succeed on step 1, 2, 3, but I fail on step 4 "mine({'cobblestone
    ':2}, null);". My inventory now has 10 planks, 4 stick.
Explainer: Because mining cobblestone needs to use the tool wooden_pickaxe.
User: Please fix the above errors and replan the task "obtain 1 stone_sword".
RePlanner: The code for crafting stone_sword is as bellows:
def obtain_1_stone_sword(inventory = {'planks':10, 'stick':4}):
   craft({'wooden_pickaxe':1}, {'planks':3, 'stick':2}, 'crafting_table'); # step
       1: craft 1 wooden_pickaxe from 3 planks and 2 stick, on crafting_table
   mine({'cobblestone':2}, 'wooden_pickaxe'); # step 2: mine 2 cobblestone with
       wooden_pickaxe
   craft({'stone_sword':1}, {'cobblestone':2, 'stick':1}, 'crafting_table'); #
       step 3: craft 1 stone_sword from 2 cobblestone and 1 stick, on
       crafting_table
   return 'stone_sword'

Descriptor: I fail on step 1 "craft({'wooden_pickaxe':1}, {'planks':3, 'stick':2},
     'crafting_table');". My inventory now has 10 planks, 4 stick.
Explainer: Because the action needs to use the tool crafting_table, but I do not
    have it.
User: Please fix the above errors and replan the task "obtain 1 stone_sword".
RePlanner: The code for crafting stone_sword is as bellows:
def obtain_1_stone_sword(inventory = {'planks':10, 'stick':4}):
   craft({'crafting_table':1}, {'planks':4}, null); # step 1: craft 1
       crafting_table from 4 planks
   craft({'wooden_pickaxe':1}, {'planks':3, 'stick':2}, 'crafting_table'); # step
       2: craft 1 wooden_pickaxe from 3 planks and 2 stick, on crafting_table
   mine({'cobblestone':2}, 'wooden_pickaxe'); # step 3: mine 2 cobblestone with
       wooden_pickaxe
   craft({'stone_sword':1}, {'cobblestone':2, 'stick':2}, 'crafting_table'); #
       step 4: craft 1 stone_sword from 2 cobblestone and 1 stick, on
       crafting_table
   return 'stone_sword'
Descriptor: I succeed in step 1, 2, 3, 4.
User: Good. I finish all steps and I obtain 1 stone_pickaxe successfully.
```

```
Planner: OK.
```

Listing 3: Prompt for DEPS in Minecraft tasks

## G.1.2   Baselines

```
User: You are a helper agent in Minecraft. You need to generate the sequences of
    goals for a certain task in Minecraft.
Planner: OK.

User: How to obtain 1 wooden_button in Minecraft step-by-step?
Planner: The plan is as follows:
step 1: mine 1 log
step 2: craft 4 planks
step 3: craft 1 wooden_button
###

User: How to obtain stone_sword in Minecraft step-by-step?
Planner: The plan is as follows:
step 1: mine 3 log
step 2: craft 12 planks
step 3: craft 4 stick
step 4: craft 1 crafting_table
step 5: craft 1 wooden_pickaxe
step 6: mine 2 cobblestone
step 7: craft 1 stone_sword
###
```

Listing 4: Prompt for Zero-shot Planner [16] in Minecraft tasks.

```
User: You are a helper agent in Minecraft. You need to generate the sequences of
    goals for a certain task in Minecraft.
Planner: OK.

User: How to obtain 1 wooden_button in Minecraft step-by-step?
Planner: The code for obtaining wooden_button is as follows:
'''
from actions import mine_log, craft_planks, craft_wooden_button
def obtain_wooden_button(inventory = {}):
   mine_log(num = 1); # step 1: mine 1 log
   craft_planks(num = 4); # step 2: craft 4 planks
   craft_wooden_button(num = 1); # step 3: craft 1 wooden_button
   return "wooden_button"
'''

User: How to obtain 1 stone_sword in Minecraft step-by-step?
Planner: The code for obtaining stone_sword is as follows:
'''
from actions import mine_log, craft_planks, craft_stick, craft_crafting_table,
    craft_wooden_pickaxe, mine_cobblestone, craft_stone_sword

def obtain_stone_sword(inventory = {}):
   mine_log(num = 3); # step 1: mine 3 log
   craft_planks(num = 12); # step 2: craft 12 planks
   craft_stick(num = 4); # step 3: craft 4 stick
   craft_crafting_table(num = 1); # step 4: craft 1 crafting_table
   craft_wooden_pickaxe(num = 1); # step 5: craft 1 wooden_pickaxe
   mine_cobblestone(num = 2); # step 6: mine 2 cobblestone
   craft_stone_sword(num = 1); # step 7: craft 1 stone_sword
   return "stone_sword"
'''
```

Listing 5: Prompt for ProgPrompt [42] in Minecraft tasks.

```
User: You are a helper agent in Minecraft. You need to generate the sequences of
    goals for a certain task in Minecraft.
Planner: OK.

User: You are a helper agent in Minecraft. You need to generate the sequences of
    goals for a certain task in Minecraft.
```

```
Planner: OK.

User: How to obtain 1 wooden_button in Minecraft step-by-step?
Planner: The plan is as follows:
step 1: mine 1 log without tool
Thinking: Inventory will add 1 log, as inventory = 1*log
step 2: craft 4 planks from 1 log
Thinking: Inventory will add 4 planks and delete 1 log, as inventory = 4*planks
step 3: craft 1 wooden_button from 1 planks
Thinking: Inventory will add 1 wooden_button and delete 1 planks, as inventory = 3*
    planks + 1*wooden_button
###

User: How to obtain stone_sword in Minecraft step-by-step?
Planner: The plan is as follows:
step 1: mine 3 log without tool
Thinking: Inventory will add 3 log, as inventory = 3*log.
step 2: craft 12 planks from 3 log
Thinking: Inventory will add 12 planks and delete 3 log, as inventory = 12*planks
step 3: craft 4 stick from 2 planks
Thinking: Inventory will add 4 stick and delete 2 planks, as inventory = 10*planks
    + 4*stick
step 4: craft 1 crafting_table from 4 planks
Thinking: Inventory will add 1 crafting_table and delete 4 planks, as inventory =
    6*planks + 4*stick + 1*crafting_table
step 5: craft 1 wooden_pickaxe from 3 planks and stick, on crafting_table
Thinking: Inventory will add 1 wooden_pickaxe, delete 3 planks and delete 2 stick,
    as inventory = 3*planks + 2*stick + 1*crafting_table + 1*wooden_pickaxe
step 6: mine 2 cobblestone with wooden_pickaxe
Thinking: Inventory will add 2 cobblestone, as inventory = 3*planks + 2*stick + 1*
    crafting_table + 1*wooden_pickaxe + 2*cobblestone
step 7: craft 1 stone_sword from 2 cobblestone and 1 stick, on crafting_table
Thinking: Inventory will add 1 stone_sword, delete 2 cobblestone, and delete 1
    stick, as inventory = 3*planks + 1*stick + 1*crafting_table + 1*wooden_pickaxe
    + 1*stone_sword
###
```

Listing 6: Prompt for Chain-of-Thought [45] in Minecraft tasks

```
User: You are a helper agent in Minecraft. You need to generate the sequences of
    goals for a certain task in Minecraft.
Planner: OK.

===============
User: Obtain 1 wooden_button in Minecraft step-by-step.

Scene: My inventory has nothing.
Planner: mine 1 log
Robot Thought: mining log needs no tool
Successful Action: True

Scene: My inventory has 1 log.
Planner: craft 4 planks
Robot Thought: crafting 4 planks needs 1 log
Successful Action: True

Scene: My inventory has 4 planks
Planner: craft 1 wooden_button
{Robot Thought}: crafting 1 wooden_button needs 1 planks
Successful Action: True

===============
User: Obtain 1 stone_sword in Minecraft step-by-step.

Scene: My inventory has nothing.
Planner: mine 3 log
Robot Thought: mining log needs no tool
Successful Action: True

Scene: My inventory has 3 log.
Planner: craft 12 planks
Robot Thought: crafting 12 planks needs 3 log
```

```
Successful Action: True

Scene: My inventory has 12 planks
Planner: craft 4 stick
Robot Thought: crafting 4 stick needs 2 planks
Successful Action: True

Scene: My inventory has 10 planks, 4 stick
Planner: craft 1 wooden_pickaxe
Robot Thought: crafting 1 wooden_pickaxe needs 3 planks and 2 stick on
    crafting_table
Successful Action: False

Scene: My inventory has 10 planks, 4 stick
Planner: craft 1 crafting_table
Robot Thought: crafting 1 crafting_table needs 4 planks
Successful Action: True

Scene: My inventory has 6 planks, 4 stick, 1 crafting_table
Planner: craft 1 wooden_pickaxe
Robot Thought: crafting 1 wooden_pickaxe needs 3 planks and 2 stick, on
    crafting_table
Successful Action: True

Scene: My inventory has 3 planks, 2 stick, 1 crafting_table, 1 wooden_pickaxe
Planner: mine 2 cobblestone
Robot Thought: mining cobblestone needs wooden_pickaxe as tool
Successful Action: True

Scene: My inventory has 3 planks, 2 stick, 1 crafting_table, 1 wooden_pickaxe, 2
    cobblestone
Planner: craft 1 stone_sword
Robot Thought: crafting 1 stone_sword needs 2 cobblestone and 1 stick, on
    crafting_table
Successful Action: True
```

Listing 7: Prompt for Inner Monologue [17] in Minecraft tasks

```
User: You are a helper agent in Minecraft. You need to generate the sequences of
    goals for a certain task in Minecraft.
Planner: OK.

User: How to obtain 1 wooden_button in Minecraft step-by-step?
Planner: The code for obtaining wooden_button is as follows:
def craft_stone_sword(inventory = {}):
   # step 1: mine 1 log without tool
   while not inventory.has({'log':1}):
      mine(obj = "log", tool = null);
   # step 2: craft 4 planks from 1 log
   while not inventory.has({'planks':4}):
      assert inventory.has({'log':1}), "I do not have enough log for crafting
          planks"
      craft(obj = "planks", tool = null);
   # step 3: craft 1 wooden_button from 1 planks
   while not inventory.has({'wooden_button':1}):
      assert inventory.has({'planks':1}), "I do not have enough planks for
          crafting wooden_button"
      craft(obj = "wooden_button", tool = null);
   return "wooden_button"
####

User: How to obtain 1 stone_sword in Minecraft step-by-step?
Planner: The code for obtaining stone_sword is as follows:
def craft_stone_sword(inventory = {}):
   # step 1: mine 3 log without tool
   while not inventory.has({'log':3}):
      mine(obj = "log", tool = null);
   # step 2: craft 12 planks from 3 log
   while not inventory.has({'planks':12}):
      assert inventory.has({'log':1}), "I do not have enough log for crafting
          planks"
      craft(obj = "planks", tool = null);
   # step 3: craft 4 stick from 2 planks
   while not inventory.has({'stick':4}):
```

```
        assert inventory.has({'planks':2}), "I do not have enough planks for
            crafting stick"
        craft(obj = "stick", tool = null);
    # step 4: craft 1 crafting_table from 4 planks
    while not inventory.has({'crafting_table':1}):
        assert inventory.has({'planks':4}), "I do not have enough planks for
            crafting crafting_table"
        craft(obj = "crafting_table", tool = null)
    # step 5: craft 1 wooden_pickaxe from 3 planks and 2 stick, on crafting_table
    while not inventory.has({'wooden_pickaxe':1}):
        assert inventory.has({'planks':3}), "I do not have enough planks for
            crafting wooden_pickaxe"
        assert inventory.has({'stick':2}), "I do not have enough stick for crafting
            wooden_pickaxe"
        assert inventory.has({'crafting_table':1}), "I do not have the tool
            crafting_table for crafting wooden_pickaxe"
        craft(obj = "crafting_table", tool = "crafting_table")
    # step 6: mine 2 cobblestone with tool wooden_pickaxe
    while not inventory.has({'cobblestone':2}):
        assert inventory.has({'wooden_pickaxe':1}), "I do not have the tool
            wooden_pickaxe for mining cobblestone"
        mine(obj = "cobblestone", tool = "wooden_pickaxe")
    # step 7: craft 1 stone_sword from 2 cobblestone and 1 stick on crafting_table
    while not inventory.has({'stone_sword':1}):
        assert inventory.has({'cobblestone':2}), "I do not have enough cobblestone
            for crafting stone_sword"
        assert inventory.has({'stick':1}), "I do not have enough stick for crafting
            stone_sword"
        assert inventory.has({'crafting_table':1}), "I do not have the tool
            crafting_table for crafting wooden_pickaxe"
        craft(obj = "stone_sword", tool = "crafting_table")
    return "stone_sword"
####
```

Listing 8: Prompt for Code as Policies [20] in Minecraft tasks

## G.2  Prompt for ALFWorld Tasks

### G.2.1  DEPS

```
Human: -= Welcome to TextWorld, ALFRED! =-
Descriptor: You are in the middle of a room. Looking quickly around you, you see a
    cabinet 1, a diningtable 1 and a fridge 1.
Human: Your task is to: put a cool tomato in cabinet. Please generate the
    sequences of sub-goals to complete the task.
Planner: The task is to put a cool tomato in cabinet. The plan is below:
    #1 pick a tomato at fridge 1
    #2 cool a tomato at fridge 1
    #3 Put a tomato in cabinet at cabinet 1
Descriptor: I fail on step 1 pick a tomato at fridge 1. I'm not carrying a tomato
    and there is no tomato, so the goal can't be accomplished.
Explainer: It's probably because there isn't a tomato at fridge 1. The task is to
    put a cool tomato in cabinet. So I should pick a tomato somewhere else. I'm
    most likely to pick a tomato at diningtable 1.
Planner: So the new plan is below:
    #1 pick a tomato at diningtable 1
    #2 put a tomato in cabinet at cabinet 1
Descriptor: I succeed on step 1.
Descriptor: I failed on step 2 put a tomato in cabinet at cabinet 1. I put a
    tomato in cabinet, but tomato 2 may not be a cool tomato, so the goal is not
    accomplished.
Explainer: It's probably because the tomato isn't cool enough. The task is to put
    a cool tomato in cabinet. So I should cool the tomato. I'm most likely to cool
    a tomato at fridge 1.
Planner: So the new plan is below:
    #2 cool a tomato at fridge 1
    #3 put a tomato in cabinet at cabinet 1
Descriptor: I succeed on step 2.
Descriptor: I succeed on step 3. I finish the task!!!

Human:-= Welcome to TextWorld, ALFRED! =-
Descriptor: You are in the middle of a room. Looking quickly around you, you see a
    desk 1, and a shelf 1.
Human: Your task is to: look at mug under the desklamp. Please generate the
    sequences of sub-goals to complete the task.
```

```
Planner: The task is to look at mug under the desklamp. The plan is below:
   #1 pick up a mug at shelf 1
   #2 use the desklamp at desk 1
Descriptor: I succeed on step 1.
Descriptor: I succeed on step 2. I finish the task!!!
```

Listing 9: Prompt for DEP in ALFWorld tasks

```
Human: You are in the middle of a room. You are not carrying anything. Your goal
    is to: pick a plate at diningtable 1.
The 5 admissible actions of the current situation are :
1. go to cabinet 1
2. go to diningtable 1
3. go to fridge 1
4. inventory
5. look
AI: The goal is to pick a plate at diningtable 1. I'm not at diningtable 1. So I
    should go to diningtable 1 first. So the answer is 2.
Human: You arrive at loc 17. On the diningtable 1, you see a plate 2, and a tomato
     2.
The 7 admissible actions of the current situation are :
1. examine diningtable 1
2. go to cabinet 1
3. go to fridge 1
4. inventory
5. look
6. take plate 2 from diningtable 1
7. take tomato 2 from diningtable 1
AI: The goal is to pick a plate at diningtable 1. I'm at diningtable 1, and there
    is a plate 2. So I should take plate 2 from diningtable 1. So the answer is 6.
Human: You pick up the plate 2 from the diningtable 1.
The 6 admissible actions of the current situation are :
1. examine diningtable 1
2. go to cabinet 1
3. go to fridge 1
4. inventory
5. look
6. put plate 2 in/on diningtable 1
AI: The goal is to pick a plate at diningtable 1. According to Human, I pick up
    the plate 2 from diningtable 1. So the goal is accomplished.

Human: You are facing fridge 1. You are carrying plate 2. Your goal is to: cool a
    plate at fridge 1.
The 6 admissible actions of the current situation are :
1. examine fridge 1
2. go to cabinet 1
3. go to diningtable 1
4. inventory
5. look
6. cool plate 2 with fridge 1
AI: The goal is to cool a plate at fridge 1. I'm at fridge 1. So I should cool
    plate 2 with fridge 1. So the answer is 6.
Human: You cool the plate 2 using the fridge 1.
The 5 admissible actions of the current situation are :
1. examine fridge 1
2. go to cabinet 1
3. go to diningtable 1
4. inventory
5. look
AI: The goal is to cool a plate at fridge 1. According to Human, I cool the plate
     2 using the fridge 1. So the goal is accomplished.

Human: You are facing diningtable 1. You are not carrying anything. Your goal is
    to: pick a plate at diningtable 1.
The 6 admissible actions of the current situation are :
1. examine diningtable 1
2. go to cabinet 1
3. go to fridge 1
4. inventory
5. look
6. take tomato 2 from diningtable 1
AI: The goal is to pick a plate at diningtable 1. I'm at diningtable 1, but there
    is no action to pick a plate. So the answer is 0(none). So the goal can't be
    accomplished, probably because there is no plate at diningtable 1.
```

```
Human: You are facing fridge 1. You are not carrying anything. Your goal is to:
    cool a plate at fridge 1.
The 5 admissible actions of the current situation are :
1. examine fridge 1
2. go to cabinet 1
3. go to diningtable 1
4. inventory
5. look
AI: The goal is to cool a plate at fridge 1. I'm at fridge 1, but there is no
    action to cool a plate. So the answer is 0(none). So the goal can't be
    accomplished, probably because I'm not carrying a plate.

Human: You are facing fridge 1. You are carrying a plate. Your goal is to: pick a
    tomato at fridge 1.
The 5 admissible actions of the current situation are :
1. examine fridge 1
2. go to cabinet 1
3. go to diningtable 1
4. inventory
5. look
AI: The goal is to pick a tomato at fridge 1. I'm at fridge 1, but there is no
    action to pick a tomato. So the answer is 0(none). So the goal can't be
    accomplished, probably because I can't pick a tomato while carrying a plate.
```

Listing 10: Prompt for the controller of DEP in ALFWorld tasks

### G.2.2 Baselines

```
Human: -= Welcome to TextWorld, ALFRED! =-
Your task is to: cool some plate and put it in cabinet.
Scene: You are in the middle of a room. Looking quickly around you, you see a
    cabinet 1, a diningtable 1 and a fridge 1.
The 5 admissible actions of the current situation are :
1. go to cabinet 1
2. go to diningtable 1
3. go to fridge 1
4. inventory
5. look
Robot Thought: The task is to cool some plate and put it in cabinet. I'm carrying
    nothing. So I need to find a plate first. There are cabinet, diningtable and
    fridge, but no plate. I'm most likely to find a plate near diningtable. So I
    should go to diningtable 1. So the answer is 2.
Robot Action: go to diningtable 1

Scene: You arrive at loc 17. On the diningtable 1, you see a plate 2, and a tomato
     2.
The 7 admissible actions of the current situation are :
1. examine diningtable 1
2. go to cabinet 1
3. go to fridge 1
4. inventory
5. look
6. take plate 2 from diningtable 1
7. take tomato 2 from diningtable 1
Robot Thought: The task is to cool some plate and put it in cabinet. I'm carrying
    nothing. So I need to find a plate first. There is a plate 2. So I should take
     plate 2 from diningtable 1. So the answer is 6.
Robot Action: take plate 2 from diningtable 1

Scene: You pick up the plate 2 from the diningtable 1.
The 7 admissible actions of the current situation are :
1. examine diningtable 1
2. examine plate 2
3. go to cabinet 1
4. go to fridge 1
5. inventory
6. look
7. put plate 2 in/on diningtable 1
Robot Thought: The task is to cool some plate and put it in cabinet. I'm carrying
    a plate 2. So I need to cool a plate first. There are cabinet, diningtable,
    and fridge. I'm most likely to cool a plate near fridge. So I should go to
    fridge 1. So the answer is 4.
Robot Action: go to fridge 1

Scene: You arrive at loc 41. The fridge 1 is closed.
```

```
The 8 admissible actions of the current situation are :
1. cool plate 2 with fridge 1
2. examine fridge 1
3. examine plate 2
4. go to cabinet 1
5. go to diningtable 1
6. inventory
7. look
8. open fridge 1
```
**Robot Thought:** The task is to cool some plate and put it in cabinet. I'm carrying a plate 2. So I need to cool a plate first. There is a fridge 1. So I should cool plate 2 with fridge 1. So the answer is 1.
**Robot Action:** cool plate 2 with fridge 1

```
Scene: You cool the plate 2 using the fridge 1.
The 8 admissible actions of the current situation are :
1. cool plate 2 with fridge 1
2. examine fridge 1
3. examine plate 2
4. go to cabinet 1
5. go to diningtable 1
6. inventory
7. look
8. open fridge 1
```
**Robot Thought:** The task is to cool some plate and put it in cabinet. I'm carrying a cool plate 2. So I need to find a cabinet first. There is a cabinet 1. So I should go to cabinet 1. So the answer is 4.
**Robot Action:** go to cabinet 1

```
Scene: You arrive at loc 20. The cabinet 1 is closed.
The 7 admissible actions of the current situation are :
1. examine cabinet 1
2. examine plate 2
3. go to diningtable 1
4. go to fridge 1
5. inventory
6. look
7. open cabinet 1
```
**Robot Thought:** The task is to cool some plate and put it in cabinet. I'm carrying a cool plate 2. So I need to find a cabinet first. There is a cabinet 1. The cabinet is closed. So I should open cabinet 1. So the answer is 7.
**Robot Action:** open cabinet 1

```
Scene: You open the cabinet 1. The cabinet 1 is open. In it, you see nothing.
The 8 admissible actions of the current situation are :
1. close cabinet 1
2. examine cabinet 1
3. examine plate 2
4. go to diningtable 1
5. go to fridge 1
6. inventory
7. look
8. put plate 2 in/on cabinet 1
```
**Robot Thought:** The task is to cool some plate and put it in cabinet. I'm carrying a cool plate 2. So I need to find a cabinet first. There is a cabinet 1. So I should put plate 2 in/on cabinet 1. So the answer is 8.
**Robot Action:** put plate 2 in/on cabinet 1

```
Scene: You put the plate 2 in/on the cabinet 1. You won!!!
```

Listing 11: Prompt for Inner Monologue [17] in ALFWorld tasks

## G.3 Prompt for Tabletop Manipulation Tasks

### G.3.1 DEPS

**system:** The template of the lang goal is as follow: put the <color> blocks in a < color> bowl. For example, "put the green blocks in a blue bowl" is a valid lang goal. Do not provide any additional explanations or instructions beyond writing lang goals.

**user:** There are 4 bowls of blue, green, pink, yellow on the table. There are 3 blocks of red, yellow, blue on the table. How to match the blocks and the bowls?

Listing 12: Prompt for DEP in CLIPort tasks

### G.3.2 Baselines

```
system: The template of the language goal is as follows: put the <color> blocks in
     a <color> bowl. For example, "put the green blocks in a blue bowl" is a valid
     lang goal. Do not provide any additional explanations or instructions beyond
     writing lang goals and determing the game is DONE. Please write a single lang
     goal in your response!

user: There are 4 bowls of blue, green, pink, yellow on the table. There are 3
     blocks of red, yellow, blue on the table. What should I do as the first step
     to match the blocks with the bowls of the literally same color? Note that not
     all the blocks can be matched and you can determine the game is DONE anywhere.
assistant: put the yellow blocks in a yellow bowl

user: OK. What should I do next?
assistant: put the blue blocks in a blue bowl

user: OK. I finish the task!
```

Listing 13: Prompt for Inner Monologue [17] in Tabletop Manipulation tasks

## H Full Dialogue

```
User: You are a helper agent in Minecraft. You need to generate the sequences of
     goals for a certain task in Minecraft.
Planner: OK.
=======================================
User: How to obtain 1 diamond in Minecraft step-by-step?
Planner: The code for task obtaining diamond is as bellows:
def obtain_1_diamond(inventory = {}):
  mine({'log':3}, null); # step 1: mine 3 log without tool
  craft({'planks':12}, {'log':3}, null); # step 2: craft 12 planks from 3 log
  craft({'stick':4}, {'planks':2}, null); # step 3: craft 4 stick from 2 planks
  craft({'crafting_table':1}, {'planks':4}, null); # step 4: craft 1
      crafting_table from 4 planks
  craft({'wooden_pickaxe':1}, {'planks':3, 'stick':2}, 'crafting_table'); # step
      5: craft 1 wooden_pickaxe from 3 planks and 2 stick, on crafting_table
  mine({'cobblestone':2}, 'wooden_pickaxe'); # step 6: mine 2 cobblestone with
      wooden_pickaxe
  craft({'stone_pickaxe':1}, {'cobblestone':3, 'stick':2}, 'crafting_table'); #
      step 7: craft 1 stone_pickaxe from 3 cobblestone and 2 stick, on
      crafting_table
  mine({'diamond':1}, 'stone_pickaxe'); # step 8: mine 1 diamond with
      stone_pickaxe
  return 'diamond'

Descriptor: I succeed on step 1, 2, 3, 4, 5, 6, but I fail on step 7 "craft({'
    stone_pickaxe':1}, {'cobblestone':3, 'stick':2}, 'crafting_table');". My
    inventory now has 7 planks, 2 stick, 1 crafting_table, 1 wooden_pickaxe, 2
    cobblestone, 5 dirt.
Explainer: Because crafting stone_pickaxe needs 3 cobblestone, but I only have 2
    cobblestone in inventory.
User: Please fix above errors and re-plan the task "obtain 1 diamond".
RePlanner: The code for task obtaining 1 diamond is as follows:
def obtain_1_diamond(inventory = {'planks':7, 'stick':2, 'crafting_table':1, '
    wooden_pickaxe':1, 'cobblestone':2, 'dirt':5}):
  mine({'cobblestone':1}, 'wooden_pickaxe'); # step 1: mine 1 cobblestone with
      wooden_pickaxe
  craft({'stone_pickaxe':1}, {'cobblestone':3, 'stick':2}, 'crafting_table'); #
      step 2: craft 1 stone_pickaxe from 3 cobblestone and 2 stick, on
      crafting_table
  mine({'diamond':1}, 'stone_pickaxe'); # step 3: mine 1 diamond with
      stone_pickaxe
```

```
    return 'diamond'
```

**Descriptor**: I succeed on step 1, 2 in new plan, but I fail on step 3 "mine({'
    diamond':1}, 'stone_pickaxe');". My inventory now has 7 planks, 1
    crafting_table, 1 wooden_pickaxe, 1 cobblestone, 5 dirt, 1 stone_pickaxe.
**Explainer**: Because mining diamond needs to use the tool iron_pickaxe, but I do not
    have it.
**User**: Please fix above errors and re-plan the task "obtain 1 diamond".
**RePlanner**: The code for obtaining 1 diamond is as follows:

```
def obtain_1_diamond(inventory = {'planks':7, 'crafting_table':1, 'wooden_pickaxe
    ':1, 'cobblestone':1, 'dirt':5, 'stone_pickaxe':1}):
  mine({'iron_ore':1}, 'stone_pickaxe'); # step 1: mine 1 iron_ore with
      stone_pickaxe
  smelt({'iron_ingot':1}, {'iron_ore':1}, 'furnace'); # step 2: craft 1
      iron_ingot from 1 iron_ore, on furnace
  craft({'iron_pickaxe':1}, {'iron_ingot':3, 'stick':2}, 'crafting_table'); #
      step 3: craft 1 iron_pickaxe from 3 iron_ingot and 2 stick, on
      crafting_table
  mine({'diamond':1}, 'iron_pickaxe'); # step 4: mine 1 diamond with iron_pickaxe
  return 'diamond'
```

**Descriptor**: I success on step 1 in new plan, but I fail on step 2 "smelt({'
    iron_ingot':1}, {'iron_ore':1}, 'furnace');". My inventory now has 7 planks, 1
     crafting_table, 1 wooden_pickaxe, 6 cobblestone, 1 iron_ore, 20 dirt, 64
    stone.
**Explainer**: Because smelting iron ingot needs to use the tool furnace, but I do not
    have it.
**User**: Please fix above errors and replan the task "obtain 1 diamond".
**RePlanner**: The code for obtaining 1 diamond is as follows:

```
def mine_1_diamond(inventory = {'planks':7, 'crafting_table':1, 'wooden_pickaxe':1,
    'cobblestone':6, 'iron_ore':1, 'dirt':20, 'stone':64}):
  craft({'furnace':1}, {'cobblestone':8}, null); # step 1: craft 1 furnace from 8
        cobblestone
  craft({'iron_ingot':1}, {'iron_ore':1}, 'furnace'); # step 2: craft 1
      iron_ingot from 1 iron_ore, on furnace
  craft({'iron_pickaxe':1}, {'iron_ingot':3, 'stick':2}, 'crafting_table'); #
      step 3: craft 1 iron_pickaxe from 3 iron_ingot and 2 stick, on
      crafting_table
  mine({'diamond':1}, 'iron_pickaxe'); # step 4: mine 1 diamond with iron_pickaxe
  return 'diamond'
```

**Descriptor**: I fail on step 1 "craft({'furnace':1}, {'cobblestone':8}, null);". My
    inventory now has 7 planks, 1 crafting_table, 1 wooden_pickaxe, 6 cobblestone,
     1 iron_ore, 20 dirt, 64 stone.
**Explainer**: Because crafting furnace needs 8 cobblestone, but I only have 6
    cobblestone.
**User**: Please fix above errors and re-plan the task "obtain 1 diamond".
**RePlanner**: The code for obtaining 1 diamond is as follows:

```
def obtain_1_diamond(inventory = {'planks':7, 'crafting_table':1, 'wooden_pickaxe
    ':1, 'cobblestone':6, 'iron_ore':1, 'dirt':20, 'stone':64}):
  mine({'cobblestone':8}, 'stone_pickaxe'); # step 1: mine 8 cobblestone with
      stone_pickaxe
  craft({'furnace':1}, {'cobblestone':8}, null); # step 2: craft 1 furnace from 8
        cobblestone
  smelt({'iron_ingot':1}, {'iron_ore':1}, 'furnace'); # step 3: craft 1
      iron_ingot from 1 iron_ore, on furnace
  craft({'iron_pickaxe':1}, {'iron_ingot':3, 'stick':2}, 'crafting_table'); #
      step 4: craft 1 iron_pickaxe from 3 iron_ingot and 2 stick, on
      crafting_table
  mine({'diamond':1}, 'iron_pickaxe'); # step 5: mine 1 diamond with iron_pickaxe
  return 'diamond'
```

**Descriptor**: I succeed on step 1 in new plan, but I fail on step 2 "craft({'furnace
    ':1}, {'cobblestone':8}, null);". My inventory now has 7 planks, 1
    crafting_table, 1 wooden_pickaxe, 1 stone_pickaxe, 8 cobblestone, 1 iron_ore,
    20 dirt, 64 stone.
**Explainer**: Because crafting furnace needs to use the tool crafting_table, but I do
     not have it.
**User**: Please fix above errors and re-plan the task "obtain 1 diamond".
**RePlanner**: The code for obtaining 1 diamond is as follows:

```
def obtain_1_diamond(inventory = {'planks':7, 'crafting_table':1, 'wooden_pickaxe
    ':1, 'cobblestone':8, 'iron_ore':1, 'dirt':20, 'stone':64}):
```

```
    craft({'furnace':1}, {'cobblestone':8}, 'crafting_table'); # step 1: craft 1
        furnace from 8 cobblestone, on crafting_table
    smelt({'iron_ingot':1}, {'iron_ore':1}, 'furnace'); # step 2: smelt 1
        iron_ingot from 1 iron_ore, on furnace
    craft({'iron_pickaxe':1}, {'iron_ingot':3, 'stick':2}, 'crafting_table'); #
        step 3: craft 1 iron_pickaxe from 3 iron_ingot and 2 stick, on
        crafting_table
    mine({'diamond':1}, 'iron_pickaxe'); # step 4: mine 1 diamond with iron_pickaxe
    return 'diamond'
```

**Descriptor:** I succeed on step 1, 2 in new plan, but I fail on step 3 "craft({'
    iron_pickaxe':1}, {'iron_ingot':3, 'stick':2}, 'crafting_table');". My
    inventory now has 6 planks, 1 crafting_table, 1 wooden_pickaxe, 1
    stone_pickaxe, 1 furnace, 1 iron_ingot, 20 dirt, 64 stone.
**Explainer:** Because crafting iron pickaxe needs 3 iron_ingot, but I only have 1
    iron_ingot.
**User:** Please fix above errors and re-plan the task "obtain 1 diamond".
**RePlanner:** The code for obtaining 1 diamond is as follows:
```
def mine_1_diamond(inventory = {'planks':6, 'crafting_table':1, 'wooden_pickaxe':1,
     'stone_pickaxe':1, 'furnace':1, 'iron_ingot':1, 'dirt':20, 'stone':64}):
    mine({'iron_ore':2}, 'stone_pickaxe'); # action 1: mine 2 iron_ore with
        stone_pickaxe
    smelt({'iron_ingot':2}, {'iron_ore':2}, 'furnace'); # action 2: craft 2
        iron_ingot from 2 iron_ore, on furnace
    craft({'iron_pickaxe':1}, {'iron_ingot':3, 'stick':2}, 'crafting_table'); #
        action 3: craft 1 iron_pickaxe from 3 iron_ingot and 2 stick, on
        crafting_table
    mine({'diamond':1}, 'iron_pickaxe'); # action 4: mine 1 diamond with
        iron_pickaxe
    return 'diamond'
```

**Descriptor:** I succeed on step 1, 2 in new plan, but I fail on step 3 "craft({'
    iron_pickaxe':1}, {'iron_ingot':3, 'stick':2}, 'crafting_table');". My
    inventory now has 5 planks, 1 crafting_table, 1 wooden_pickaxe, 1
    stone_pickaxe, 1 furnace, 3 iron_ingot, 20 dirt, 64 stone.
**Explainer:** Because crafting iron_pickaxe needs 2 stick, but I do not have it.
**User:** Please fix above errors and re-plan the task "obtain 1 diamond".
**RePlanner:** The code for obtaining 1 diamond is as follows:
```
def obtain_1_diamond(inventory = {'planks':5, 'crafting_table':1, 'wooden_pickaxe
    ':1, 'stone_pickaxe':1, 'furnace':1, 'iron_ingot':3, 'dirt':20, 'stone':64}):
    craft({'stick':2}, {'planks':1}, null); # action 15: craft 2 stick from 1
        planks
    craft({'iron_pickaxe':1}, {'iron_ingot':3, 'stick':2}, 'crafting_table'); #
        action 16: craft 1 iron_pickaxe from 3 iron_ingot and 2 stick, on
        crafting_table
    mine({'diamond':1}, 'iron_pickaxe'); # action 17: mine 1 diamond with
        iron_pickaxe
    return 'diamond'
```
**Descriptor:** I succeed in step 1, 2, 3.
**User:** Good. I finish all steps and I obtain 1 stone_pickaxe successfully.
**Planner:** OK.

Listing 14: Full dialogue of task `ObtainDiamond`

