# OpenReview forum: "Describe, Explain, Plan and Select: Interactive Planning with LLMs Enables Open-World Multi-Task Agents"
_NeurIPS.cc/2023/Conference — NeurIPS 2023 poster_

### Official Review · Reviewer_xCHc · 2023-06-25

**Soundness:** 3 good
**Presentation:** 4 excellent
**Contribution:** 2 fair
**Rating:** 6
**Confidence:** 5

**Summary:**

This paper focuses on building an LLM-based planner for sequential decision-making tasks in open-world environments such as Minecraft. It proposes a method called "Describe, Explain, Plan and Select" (DEPS), which incorporates multiple stages including LLM describe, LLM explain, planning, and goal selection. The system is evaluated on hundreds of tasks in Minecraft. Several baseline methods are also compared against.

**Strengths:**

- Clear motivation. The proposed method is well motivated from the angles from unique challenges associated with open worlds.
- Excellent presentation. The paper is overall well presented, with coherent flow and easy-to-read figures that explain key concepts in the method.
- Thorough evaluation and comparisons. 5 methods proposed in prior work are discussed and compared against in the experiment section.
- Code is included for reproducibility, although I didn't run it.

**Weaknesses:**

# Novelty of the proposed method is limited to certain extent
As illustrated in Figure 7 in appendix which compares the proposed method against existing methods, the proposed method only introduces three extra modules, namely the "Explainer", "Descriptor", and "Selector". However, their significance or importance are not well justified as explained below.

## On Explainer
At L58, the role of Explainer is to locate the errors in the previous plans. However, according to example prompt provided to the Explainer as listed at L523-L549, it is trivial to summarize the errors because necessary information is already provided and well structured. What the Explainer does is merely compare what items are missing given the recipe, and rephrase it to sentences like "[...] so I need to craft/mine ... first". This makes the Explainer's necessity inconceivable.

## On Descriptor
While the paper claims the Descriptor to be models such as VLMs (Figure 2 and L120), in fact, it is merely rule-based conversion that maps symbolic environment state information to language (L151-152). Table 9 further provides more evidence to this. Since environments considered in this work usually provide rich privileged information that is critical to accomplish corresponding tasks, it is unclear that whether performance improvements come from the introduced module or simply the access to privileged information.

## On Selector
What is the exact setting used in Figure 4 to verify the effectiveness of Selector? At the current version the setup seems to be too easy to be representative (plain biome with 4 similar goals). The non-trivial performance of the random goal selection baseline also suggests this. Further analysis and insights into why the Selector works would be helpful. It's also good to know performance of the learned goal selector. For example, if you represent the remaining horizon as continuous value, what is the L1 distance between the prediction and ground-truth?

# Potential unfair evaluation and comparisons
## On main results
In Table 2 it’s clear that success rates monotonically improve with the number of rounds. L275-276 says that DEPS uses the maximum possible number of rounds. So it could be the case that baselines are not given the same amount of LLM tokens consumed. Clarification on this will increase the transparency of the main experiment.

## On the obtain diamond challenge
Since DEPS is based on a powerful LLM that is highly likely to be pre-trained on Minecraft-related corpus, it is not directly comparable to methods mentioned in section 4.4 that are learned without such built-in knowledge. More appropriate comparisons would be conducted against other concurrent LLM-based Minecraft systems such as GITM [Zhu et al., 2023] and Voyager [Wang et al., 2023]. Including the discussion about them is also encouraged. Additionally, VPT's performance is wrongly reported. It should be around 20% for collecting diamonds and 2.5% for collecting diamond pickaxe.

# Highly variant performance for DEPS on challenging tasks
Performance of the proposed method is highly variant on challenging tasks, specifically, starting from MT7, where the one-sigma range includes negative success rate. Further statistical analysis or experiments to justify that the task success is not from randomness will be appreciated.

# Inappropriate word choice
At L63 the system is described as "zero-shot". Analogy to Brown et al., 2020 is drawn to justify the word choice. However, if we follow the same naming convention in Brown et al., 2020, the proposed system would be better described as "few-shot" since multiple task-related demonstrations are provided in prompts. Similarly, at L129, "one-shot planning methods" should be better called as "open-loop planning methods" given the context that there is no re-planning involved in those methods.

Overall, I appreciate the efforts put in this work. I'm happy to raise the score if concerns/questions above can be addressed.

- Zhu et al., Ghost in the Minecraft: Generally Capable Agents for Open-World Environments via Large Language Models with Text-based Knowledge and Memory, 2023.
- Wang et al., Voyager: An Open-Ended Embodied Agent with Large Language Models, 2023.
- Brown et al., Language models are few-shot learners, 2020.

**Questions:**

See Weakness section

**Limitations:**

See Weakness section

---

> ### Author Rebuttal · Authors · 2023-08-10
>
> > Concerns on Explainer
> >
>
> Regarding the example mentioned by the reviewer (L523-L549), the explanation generated by the Explainer contains the potential reason for the current failure, not just summarizing the current state. Specifically, the LLM is prompted to explain the reason for the failure using its own knowledge of Minecraft. For example, in lines 528-531, "*So I need to craft stone_pickaxe first*" **cannot be concluded directly from the environment state using rules, it can only be derived through LLM's knowledge.**
>
> In our experiments, we observe that the re-planning success rate decreases significantly without the Explainer, even when direct environmental feedback is given. We conducted ablation experiments in which we removed the Explainer based on DEP in 71 categories of Minecraft tasks. The results showed that compared to DEP, the average success rate dropped from 39.36 (DEP) to 30.25 (DEP w/o Explainer).
>
> We can provide concrete qualitative examples of DEP and DEP w/o Explainer but not showing here due to space limit. Feel free to reach out if you want to see any during the discussion period.
>
> > Concern on Descriptor.
> >
>
> The descriptor is a hybrid system. The inputs we use include 1) inventory, which is directly converted; 2) images, which can be converted with the help of VLM; 3) biome, compass, and location, which are also directly converted. In addition to these, we do not utilize any other information. Therefore, the descriptor does not use privileged information (e.g., Lidar or Block information) from the environment and all the information it uses is what the player can actually access.
>
> > Concern on Selector.
> >
>
> Thanks for raising this. We commit to including more details in the final version. Here, we explain the setting of experiments on the selector for your reference:
>
> In the ablation experiments, we put both agents and some targets (sheep, cow, pig, and a tree) in the Plains biome. The agent is tasked to complete a various number of goals (Hunt a sheep/cow/pig and Mine a log) within **a limited period of time (the x-axis of Fig.4)**. These goals can be completed in any order, but the distance from the agent to these targets is different. Therefore, an optimal plan, which takes minimal time steps, will be to complete the goal in an order based on the proximity to the agent. Our selector is introduced to help with this. We then try obtaining an order from different selectors (Horizon-predictive or HPS, CLIP, MineCLIP, Random or Fixed) and complete the goals following it. Obviously, a better selector can lead to more optimal order, saves more time, and results in a higher success rate (the y-axis of Fig.4). Please find the numbers (success rate and average number of task completion) across different selector in Table 2&3 of the global rebuttal PDF. They further demonstrate the effectiveness of the horizon-based selector we proposed.
>
> As you can see, this is a meticulously designed experiment setting that precisely measures how well selectors help with DEPS. We agree more realistic setting could be more convincing, therefore we refer you to the main result in Table 1 (of the main paper): DEP vs DEPS, where DEP + selector (DEPS) shows clear advantages over DEP.
>
> Lastly, we will include more direct prob of the selector, such as how well it predicts the goal proximity as part of future work. At this point, we focus on showing the benefits it brings to DEPS on task completions, as we already mentioned. Please let me know if this addresses your concern about the selector.
>
> > Concern on main results.
> >
>
> Some baseline methods (like Chain-of-Thought and Huang et al) cannot re-plan, so more rounds are meaningless to them. Other methods that can perform re-planning, e.g. Inner Monologue, also use the maximum possible number of rounds as DEPS does.
>
> > Concern on obtain diamond challenge.
> >
>
> In the process of reproducing VPT (rl-from-early-game-2x), we only successfully mined diamonds once in 40 attempts. Therefore, the probability of obtaining diamonds by VPT under the same setting is reported in the paper. We will modify the success rate of obtaining diamonds by VPT to 20% in the next version of the article and add our own reproduction in a footnote.
>
> Compared to GITM and Voyager calling the API for low-level control, DEPS uses visual observation and mouse-keyboard action space, so the experimental settings are totally different and are not comparable at all.
>
> We noticed that in order to make the environment simpler, GITM and Voyager modified the game and used a lot of privileged information from the environment, turning the game into text-based. In addition, GITM did not release the code, making it difficult to evaluate its success rate in our unmodified Minecraft environment (without strength and other effects).
>
> Finally, GITM and Voyager are works submitted to arXiv **after the NeurIPS deadline**.
>
> > Concern on DEPS variant performance.
> >
>
> We understand your concern about variance. We tested each method with at least 30 different seeds, resulting in completely different Minecraft landscapes across trials and high variance. But we believe our results are still significant. Here we provide the average p_value of DEPS vs CaP(the best baseline) of the results in Table 1(main paper) to justify our claim: p_value = 6.3076e-05. As you can see, our results are very significant (w/ p_value < 0.005). Let us know if you still have concerns and we’re happy to assist further.
>
> > Concern on word choice.
> >
>
> There is no consensus in the community on this issue. For naming, we referred to Huang et al and Socratic models and call our approach "zero-shot". We understand your concern and we will explain it more clearly in the article.
>
> - Huang et al. Language models as zero-shot planners: Extracting actionable knowledge for embodied agents
> - Socratic models: Composing zero-shot multimodal reasoning with language

---

> > ### Comment · Reviewer_xCHc · 2023-08-15
> >
> > Thank you authors for answering my questions. Could authors further explain positional arguments used in primitive functions such as "craft"? I'd also encourage authors to include ablation results of DEP w/o Explainer in the appendix in future version.

---

> > > ### Author Response · Authors · 2023-08-15
> > >
> > > We thank you for your comments. We will explain the two primitive functions in “Prompt 1” of the main paper. Before we start, we would like to first illustrate how DEPS interacts with the low-level controller. Given a task, DEPS will ultimately produce a plan, which is effectively some Python code like in “Prompt 1”. Assume the plan is correct, then every primitive function in the Python code will be sent to a parser, which will **only extract the task information needed by our low-level controller** (we will explain what information is needed below). Finally, the controller receives the task information and executes the task. Once finished (and successful), we will move to the next primitive function and repeat this process.
> > >
> > > Now we explain the two primitive functions:
> > >
> > > `mine(obj, tool)`: only the name `mine` and the 1st positional argument `obj` are needed by the controller. The controller will automatically choose the right tool and try to mine the specified object. Please note: by design, **the 2nd argument `tool` is added only to help with LLM planning** (we think this is some kind of "chain-of-thought") but will **not** be used by the low-level controller.
> > >
> > > `craft(obj, materials, tools)`: only the name `mine` and the 1st positional argument `obj` are needed by the controller. The controller will automatically use the necessary materials and tools to craft the specified object. The reason why we have the 2nd and 3rd positional argument is the same as in `mine(obj, tool)` -- introducing some sort of "chain-of-thought" prompting.
> > >
> > > We will commit to including the additional ablation results of DEP w/o Explainer in the appendix in future version.

---

> > > > ### Comment · Reviewer_xCHc · 2023-08-15
> > > > **Thanks for clarification**
> > > >
> > > > I appreciate authors' efforts to answer my further questions. I would encourage authors to include clarifications happened during the discussion into the final version. I am happy to vote for acceptance and changed my score.

---

### Official Review · Reviewer_W5LM · 2023-07-05

**Soundness:** 3 good
**Presentation:** 3 good
**Contribution:** 3 good
**Rating:** 5
**Confidence:** 5

**Summary:**

The paper discusses the problem of planning in Minecraft and proposes an interactive planning approach based on Large Language Models (LLMs) called DEPS. The LLM based planner has the ability to re-plan upon failure and can take distance into consideration.

**Strengths:**

1. To my knowledge, this is one of the first works showing a working embodied agent with LLM planner.
2. The DEPS architecture is quite reasonable and intuitive, and achieves promising results.
3. I think this would lead to an exciting direction to work on. Given the popularity and challenge of Minecraft, I believe LLM solutions might achieve generalizable success in other domains.
4. The evaluation metrics are well-defined, although I hope the authors could link better back to prior works.

[After discussion/rebuttal]
+ This work is conducted relatively early with codex, which has weaker abilities compared to the latest LLMs. There could be even more potential for improvement.
+ This work should be reasonably reproducible.

**Weaknesses:**

1. It is quite confusing that the long standing ObtainDiamond challenge has been re-labeled as MT8 in a lot of result tables
2. The baselines for comparison in the tables do not seem so convincing. I wonder why VPT results are not included in table 8.

[After rebuttal]
1 has been resolved.
2 has mostly been resolved.

Some weakness remains of concern:
+ The authors were not able to reproduce VPT performance to full extent. From my knowledge, the training of the controller actually depends on demonstration collected from VPT.
+ One additional concern is the dependence on prior-works/code-bases might make the model hard to adapt/work on. Cai et al.

**Questions:**

Overall, I think this work offers a good contribution, but I hope that the authors would address some questions and concerns for me to change my score:

1. Is it possible to include better baselines, and more details on how the low-level agent is trained/made? From my understanding, this model was built on MineAgent or VPT as low-level agent, but I don't think this was mentioned explicitly.
2. It seems from table 1 that the agent still struggles to get to diamond. What do you think is the main outstanding difficulty?

[After rebuttal]
All questions have been resolved

**Limitations:**

See Question 1 for limitations.
Social impacts are addressed adequately.

[After rebuttal]
All concerns have been addressed

---

> ### Author Rebuttal · Authors · 2023-08-10
>
> Thanks for your comments. We provide clarifications below on the major comments, which will be included in the final version.
> > It is quite confusing that the long standing ObtainDiamond challenge has been re-labeled as MT8 in a lot of result tables
> >
>
> Thanks for your comments. We will further clarify this point in the modified paper.
>
> Our agent can complete over 70 tasks in Minecraft with zero-shot learning. For demonstration purposes, we have divided these 70 tasks into 8 groups based on their difficulty, namely MT1-MT8. Among them, ObtainDiamond is classified as the most challenging task and belongs to MT8. We provide a detailed description of each group's task settings in Appendix Table-8. There are other diamond-related tasks in MT8 (e.g., ObtainDiamondHoe, ObtainDiamondPickaxe etc.), and we will report more in the future.
>
> > The baselines for comparison in the tables do not seem so convincing. I wonder why VPT results are not included in table 8.
> >
>
> First of all, please note table 8 (in the appendix) only lists all our tasks but no results. We assume you mean table 1 in the main paper. Please let us know if this is a misunderstanding.
>
> VPT is rather an intrinsic rather than a goal-following agent. So in multi-task settings, what task VPT will complete is not controllable. To let VPT complete a certain type of task, it has to be fine-tuned. And once it is fine-tuned, it is no longer a multi-task agent. Therefore we do not include VPT in table 1.
>
> > Is it possible to include better baselines, and more details on how the low-level agent is trained/made? From my understanding, this model was built on MineAgent or VPT as low-level agent, but I don't think this was mentioned explicitly.
> >
>
> DEPS primarily focuses on generating more feasible plans for long-horizon tasks in an open-world scenario, using a pretrained language-conditioned controller. **We already put the details of the controller in Section-C of the Appendix. Specifically, it is built upon Cai et al.** We focus on how to implement a multi-task agent in Minecraft, a setting that prior works have not considered. As a result, all the baselines presented in this paper represent our best efforts to adapt prompts for use in Minecraft. Recently, there have also been others considering similar work as us, such as Voyager and Ghost in Minecraft, but they do not have a low-level controller that uses the same visual image observation and action space as humans. Rather, they rely on the API provided by Minecraft for low-level control, which is also unfair to compare directly.
>
> - Cai et al. Open-world multi-task control through goal-aware representation learning and adaptive horizon prediction. CVPR 2023.
> - Voyager: An Open-Ended Embodied Agent with Large Language Models. Arxiv:2305.16291
> - Ghost in the Minecraft: Generally Capable Agents for Open-World Environments via Large Language Models with Text-based Knowledge and Memory. Arxiv:2305.17144
>
> > It seems from table 1 that the agent still struggles to get to diamond. What do you think is the main outstanding difficulty?
> >
>
> As task difficulty increases (ex. ObtainDiamond), DEPS's success rate gradually declines. This can be analyzed from two perspectives: planning and control. Details are as follows. We will commit to including such discussions in the final version.
>
> Firstly, harder tasks like ObtainDiamond typically require more reasoning steps and re-planning rounds (as shown in Table-2), necessitating longer context length. We found LLMs will struggle to catch the needed information when the context becomes longer, even with powerful GPT-3 and ChatGPT, which restricts their zero-shot planning ability for harder tasks.
>
> Secondly, harder tasks often involve mining rare ores that are naturally takes a lot of time for the low-level controller to obtain in Minecraft. For instance, when it comes to diamond mining, human players have probabilities of obtaining log, cobblestone, iron ingot, and diamond within 10 minutes at rates of approximately 99%, 99%, 60%, and 15% respectively. DEPS exhibits a similar decreasing trend in this regard as well, largely due to the raising low-level controller failures as the game time is limited. Therefore, increasing the game time allotted for harder tasks may help alleviate this downward trend.
>
> Final note: some concurrent work claims a higher success rate in obtaining diamond (ex. Ghost in the Minecraft). **However, they do not incorporate low-level control or modify the game engine to simplify the task, both of which do not happen in our setting — we stick to the original Minecraft game with no modification.**

---

> > ### Comment · Reviewer_W5LM · 2023-08-10
> >
> > Thank you for the detailed answers to my questions.
> >
> > I still find it quite unclear how your low-level controller is trained even after looking at the appendix. Do you mind attaching more documentation on those implementation/training details in this paper? I think it is quite an important aspect for understanding the algorithm, but it was not covered sufficiently.
> >
> > Here are some additional questions:
> >
> > Q1 By comparing VPT results, I understand that VPT is not a controllable agent. But if you just run VPT with its trained goal to obtain diamond, how well does it do in completing the intermediate goals? (Although it is not following the specific goals, it should finish a lot of intermediate goals to diamond due to the game structure.)
> >
> > Q2 ``harder tasks like ObtainDiamond typically require more reasoning steps and re-planning rounds''. Do you mind explaining this more or giving some examples? I think this is quite important for us to understand the mechanism.

---

> > > ### Author Response · Authors · 2023-08-12
> > >
> > > Thanks for the follow-up discussion!
> > >
> > > > low-level controller details
> > >
> > > Thanks for your comments. DEPS directly uses the controller from Cai et al [1], except that the language goal set has been expanded from 10 goals to over 100 goals. We have listed some necessary details below:
> > >
> > > - Model: The Controller uses the same structure as Cai et al, which is a transformer architecture. To make the controller better following our language goal, we employ a clip text encoder to encode language text into embeddings and incorporate the Goal-Sensitive Backbone (GSB) from Cai et al for improved state representations. For more information on the model structure, refer to Figure 3 in Cai et al's work.
> > > - Training: The Controller is optimized with imitation learning on the offline dataset provided by Cai et al. It is worth noting that the language goal-conditioned trajectory trained by the controller is usually a short-term goal, such as mining log or mining cobblestone with a wooden_pickaxe.
> > > - Training data collection: The collection of training data relies on VPT. We provide various tools to VPT, allowing it to freely explore the environment and save the trajectory of successfully obtaining target objects as training data. This trajectory is labeled as collected items. For items that VPT fails to collect, we use human player game trajectories as training data. The detailed training data collection pipeline can be found in Appendix A.2 in Cai et al [1].
> > >
> > > In the process of DEPS reasoning, we did not provide any initial materials to the agent. In other words, DEPS needs to combine various skills on its own to complete the task of obtaining cobblestone. Specifically, in obtaining cobblestone, the controller skills that DEPS needs to use are: mine log, craft planks from log, craft stick from planks, craft crafting_table from planks, craft wooden_pickaxe from planks and stick, mine cobblestone with wooden_pickaxe.
> > >
> > > [1] Cai et al. Open-World Multi-Task Control Through Goal-Aware Representation Learning and Adaptive Horizon Prediction. CVPR 2023.
> > >
> > > > Q1: VPT scores
> > >
> > > In the original VPT paper, there is indeed success rate report of completing some intermediate goals. In our early experiments, we also follow their instructions to reproduce the results. But there is a discrepancy, shown below:
> > >
> > > |  | log | planks | stick | crafting_table | wooden_pickaxe | furnace | stone_pickaxe | iron_ore | iron_pickaxe | diamond |
> > > | --- | --- | --- | --- | --- | --- | --- | --- | --- | --- | --- |
> > > | VPT (original) | 1.0 | 1.0 | 1.0 | 1.0 | 0.99 | 0.99 | 0.99 | 0.99 | 0.88 | 0.2 |
> > > | VPT (reproduced) | 1.0 | 1.0 | 1.0 | 0.925 | 0.625 | 0.15 | 0.125 | 0.075 | 0.025 | 0.025 |
> > >
> > > The VPT model being used is `rl-from-early-game-2x`, evaluated for 40 episodes, 10 minutes each. We've been in contact with the authors of VPT to help resolve such discrepancy but unfortunately, their suggestions didn't work well on our side. We commit to communicating with them and keep everyone updated. In any case, we hope the results above help you understand the actual performances of VPT in completing these tasks.
> > >
> > > Please note, this is the best we can do on evaluating VPT in a multi-task setting, but it is still **quite far** from what a true goal-conditioned multi-task agent like DEPS can do — our benchmark tasks MT1-8 are much wider (and could be more complex) than obtaining the intermediate gradients in the table above. Short answer: we don’t report VPT in Table 1 (benchmarking multi-task agents on MT-1-8) because it does not have such capacity.
> > >
> > > Feel free to let us know if you have further questions.

---

> > > > ### Author Response · Authors · 2023-08-12
> > > >
> > > > > Q2: Explanation on re-planning rounds
> > > >
> > > > A more “complex” task in the context of our paper means achieving the goal of this task, ex. obtaining a diamond, requires crafting more intermediate tools. Since each of these tools is a subgoal, the final plan with become much more complicated and therefore needs more extensive reasoning. Also, the planner can be more vulnerable to mistakes, which will lead to more re-planning.
> > > >
> > > > Such re-planning rounds are at the heart of the **interactive planning** of DEPS. In **Appendix Section H**, we provide a “full dialogue” of how DEPS interactively plans, explains, understands its own mistakes, and replans to achieve a relatively complex task “obtain one diamond”. We encourage you to read that dialogue, which illustrates how replanning works in DEPS.
> > > >
> > > > To make the response more self-contained, here we also provide a simplified re-planning scenario of a task “obtain one stone_pickaxe”:
> > > >
> > > > First, DEPS is asked to generate a plan to “obtain one diamond”, here is the initial plan it produces:
> > > >
> > > > ```jsx
> > > > 1. mine 2 log
> > > > 2. craft 8 planks from 2 log
> > > > 3. craft 4 stick from 2 planks
> > > > 4. craft 1 crafting_table from 4 planks
> > > > 5. mine 3 cobblestone
> > > > 6. craft 1 stone_pickaxe from 3 cobblestone and 2 stick on crafting_table
> > > > ```
> > > >
> > > > Each goal in the plan will be passed to the Controller for execution. The Controller can successfully complete step 1 to 4, but it fails at step 5 because there is an error in the plan — according to Minecraft's game rules, to obtain cobblestone, you need to use at least a wooden_pickaxe. However, this action of crafting a wooden_pickaxe is not in the plan. Therefore, Descriptor will describe that Agent fails when executing step 5 and Explainer will analyze that the reason for Agent's failure is due to lacking a wooden_pickaxe based on Agent's current condition. Then based off all these, DEPS Re-planner will generate a new plan:
> > > >
> > > > ```jsx
> > > > 1. craft 1 wooden_pickaxe from 3 planks and 2 stick
> > > > 2. mine 3 cobblestone with wooden_pickaxe
> > > > 3. craft 1 stone_pickaxe from 3 cobblestone and 2 stick on crafting_table
> > > > ```
> > > >
> > > > The regenerated plan is again handed over to the controller for step-by-step execution. However, the controller will fail on step 1 "craft 1 wooden_pickaxe". The Explainer will provide the reason: the current agent only has 2 planks, which does not meet the required number of planks for crafting a wooden_pickaxe. In the end, the Re-planner will provide a new plan:
> > > >
> > > > ```jsx
> > > > 1. mine 1 log
> > > > 2. craft 4 planks from 1 log
> > > > 3. craft 1 wooden_pickaxe from 3 planks and 2 stick on crafting_table
> > > > 4. mine 3 cobblestone with wooden_pickaxe
> > > > 5. craft 1 stone_pickaxe from 3 cobblestone and 2 stick on crafting_table
> > > > ```
> > > >
> > > > Assuming that the controller can successfully complete each step, DEPS is able to ultimately complete the task of 'obtaining stone_pickaxe from scratch'. In other words, DEPS successfully completed this task by re-planning for two rounds (generating three plans in total). As the complexity of the task increases, the initial plan usually contains more errors, thus requiring more re-planning rounds to generate a final executable plan.
> > > >
> > > > Final note: Performing interactive re-planning is indeed a crucial contribution made by DEPS, we also conduct ablation experiments for re-planning rounds in **Section 4.3.2**. Feel free to take a look.
> > > >
> > > > Thanks for your discussion. Feel free to let us know if you have further questions.

---

> > > > > ### Comment · Reviewer_W5LM · 2023-08-13
> > > > >
> > > > > Thank you for the detailed answer to my questions. My concerns have been largely addressed, and I have updated my scores accordingly.
> > > > > [Edit: Specifically, I changed my score (4->5)]
> > > > >
> > > > > I have the following requests for the final version of the paper:
> > > > > 1. Please offer self-contained information on the training mechanism in the appendix instead of simply referring to prior work.
> > > > > 2. Please put DEPS success rate alongside VPT scores in the VPT (original vs. reproduced) table, and include them in the appendix.
> > > > > 3. Please include some failure examples in-which DEPS fails to recover.

---

> > > > > > ### Author Response · Authors · 2023-08-14
> > > > > >
> > > > > > Thanks for your very encouraging reply! We're happy to see our rebuttal has addressed your concern. We will commit to including what you have suggested: 1) more information on controller training; 2) DEPS vs VPT; 3) failure examples for replanning, in the final version.

---

> > > > > > > ### Comment · Reviewer_FjP3 · 2023-08-16
> > > > > > >
> > > > > > > This is not part of my review, just a note: I also tried running the `rl-from-early-game` VPT model to see the statistics, and over 70 episodes (same settings as reported in VPT paper), I was able to get diamond in 14 of the episodes (19.44%) and diamond pickaxe once (1.39%).
> > > > > > >
> > > > > > > Here are the steps I did:
> > > > > > > - Installed the exact versions of Python packages reported in the VPT repository (`pip install -r requirements.txt´) in a fresh Python environment. Pytorch was set to `1.9.0+cu111`.
> > > > > > > - Added manual 10min timeout  to the `run_agent.py` code (maximum of 20 * 60 * 10 steps per episode, after which we reset)
> > > > > > > - I roll out the model and store the following entries from the _final_ observation per episode: "mine_block", "pickup", "craft_item". These tell total episode statistics of blocks mined, items picked up and items crafted.
> > > > > > >
> > > > > > > This does not influence my review in any way, I just got curious when heard your results. I understand these are very finnicky things to setup, and a minute change in underlying libraries may throw off the results (e.g., image is preprocessed in slightly different ways). As long as you report your results and the disparency, I feel this is sufficient. Since your controllers were trained using data from VPT models, they should get better (if anything) with a more competent VPT agent.

---

> > > > > > > > ### Author Response · Authors · 2023-08-16
> > > > > > > >
> > > > > > > > Thanks for your comments.
> > > > > > > >
> > > > > > > > We did encounter some difficulties in reproducing VPT. First of all, it needs to be clarified that all our methods run in minedojo [1], while VPT relies in minerl@1.0 [2]. Although they both run on the Minecraft simulator, minedojo depends on Minecraft 1.11 while MineRL@1.0 depends on Minecraft 1.16.
> > > > > > > >
> > > > > > > > To ensure fairness in comparison, we first attempted to reproduce VPT in minedojo but found differences in the action space between minedojo and minerl. Specifically, mouse movement operations were canceled in minedojo and item crafting was done through a special action [3]. This made it difficult for VPT to run smoothly in minedojo, resulting in significant performance impact.
> > > > > > > >
> > > > > > > > To further test the performance of VPT, we also tested it in minerl. To reduce environmental differences, we set the resolution uniformly to 224x224 and the brightness gamma value to 1. We conducted 40 rounds of testing in this environment and found that VPT only had a 2.5% chance of obtaining diamonds.
> > > > > > > >
> > > > > > > > We reported these results to the authors of VPT and had further discussions with them. After comparing their settings, we discovered that the reason for the decreased performance of VPT lies in the different resolutions and brightness gamma values used for Minecraft observations. During rl finetuning process, VPT uses fixed environment parameters with a resolution of 640x480 and gamma of 2. Although VPT scales down observation images before inputting them into networks (128x128), differences in rendering resolutions still significantly affect `rl-from-early-game` performance.
> > > > > > > >
> > > > > > > > After setting the resolution and gamma value back to default settings, during a test consisting of 64 rounds lasting for 10 minutes each round, VPT achieved success rates consistent with those mentioned in the original paper:18.75% for obtaining diamonds and diamond_pickaxe at a rate of 1.5%. We are also conducting further investigations into more environmental details to see if there are any other settings that may affect VPT's performance. We will update our promised comparison between VPT and ours in the final version. Please note: since our benchmark tasks (MT1-8) cover a wide range of tasks and many of them cannot be accomplished during the journey of mining a diamond (this is basically what VPT is doing), it is expected VPT will still fall short than some baselines and DEPS, even with this improved result.
> > > > > > > >
> > > > > > > > Additionally, considering that the action space limitations of minedojo environment hinder the performance of VPT, we have recently been trying to collect data and train controllers in minerl instead. We hope to further improve DEPS' performance and we will add DEPS' performance in minerl in the final version as well.
> > > > > > > >
> > > > > > > > A final note in case you're interested in some empirical observations of VPT: we noticed that compared to `rl-from-early-game-2x`, `foundation-model-2x` is less sensitive to environmental settings such as resolution.
> > > > > > > >
> > > > > > > > [1] https://github.com/MineDojo/MineDojo
> > > > > > > >
> > > > > > > > [2] https://github.com/minerllabs/minerl/tree/v1.0.0
> > > > > > > >
> > > > > > > > [3] https://docs.minedojo.org/sections/core_api/action_space.html

---

> > > > > > > > > ### Author Response · Authors · 2023-08-21
> > > > > > > > >
> > > > > > > > > DEPS primarily focuses on generating more feasible plans for long-horizon tasks in an open-world scenario, and the plan will then be executed step by step by low-level goal-conditioned policies. Therefore, the overall performances of our agent (DEPS for planning and goal-conditioned policies for control) can be affected by the low-level control policies. In our experiments, we’ve observed that the environment (Minedojo vs. MineRL v1.0) does have an impact on the controller, and ultimately leads to performance discrepancy of our agent.
> > > > > > > > >
> > > > > > > > > Currently, the results we reported in the main paper and the rebuttal above are obtained from Minedojo. Based on your feedback (we thank you for that!), here we provide some preliminary results on MineRL v1.0, which is aligned with VPT. Please note that:
> > > > > > > > >
> > > > > > > > > - The goal-conditioned policies are re-trained on MineRLv1.0 as well. Thanks to the contractor data (demonstrations) offered by VPT, we’re able to train the policies with more data.
> > > > > > > > > - Due to limited time, each task was only tested with 10 seeds.
> > > > > > > > >
> > > > > > > > > | task | success rate on MineRL v1.0 | success rate on Minedojo | time limit |
> > > > > > > > > | --- | --- | --- | --- |
> > > > > > > > > | obtain_chest | 90% | 76.7% | 3 min |
> > > > > > > > > | obtain_crafting_table | 90% | 90.0% | 3 min |
> > > > > > > > > | obtain_wooden_pickaxe | 80% | 80.0% | 3 min |
> > > > > > > > > | obtain_lever | 80% | 83.3% | 6 min |
> > > > > > > > > | obtain_stone_pickaxe | 70% | 73.3% | 6 min |
> > > > > > > > > | obtain_furnace | 60% | 66.7% | 6 min |
> > > > > > > > > | obtain_iron_shovel | 30% | 13.3% | 10 min |
> > > > > > > > > | obtain_iron_pickaxe | 20% | 10.0% | 10 min |
> > > > > > > > >
> > > > > > > > > We made the following observations from the results:
> > > > > > > > >
> > > > > > > > > - The environment does have some impact on the performances, but our agent is able to still achieve good scores on the new MineRL v1.0 environment.
> > > > > > > > > - By providing more training data, our agent can be much better on some tasks, especially those that require longer steps to complete, ex. `obtain_iron_shovel`, and `obtain_iron_pickaxe`.
> > > > > > > > >
> > > > > > > > > We commit to adding complete experimental results in the final version.

---

### Official Review · Reviewer_FjP3 · 2023-07-06

**Soundness:** 3 good
**Presentation:** 4 excellent
**Contribution:** 3 good
**Rating:** 9
**Confidence:** 5

**Summary:**

The paper proposes a new planning algorithm for solving tasks in open-ended environments, and tests its performance against baselines in Minecraft. The proposed model uses large language models (LLMs) to split the large task into subgoals (for which there are controllers that can complete the subgoals). However, this type of planning often fails, as shown in the results. In addition to the planner, the proposed algorithm summarizes errors when they occur and then update the plan, which improves the results. Finally, the method includes a selector module, which is trained to select the quickest goal out of the available ones, which further improves results, especially in long-term tasks.

**Strengths:**

- Good explanation of the tackled challenge. Figures 1 and 2 are especially descriptive.
- Good coverage of the baselines and ablations to study what changes matter.
- The controller models used in the experiments acted based on image pixels and surrounding voxels, demonstrating that the proposed method can be used with such sub-policies instead of code.
- Proposed method is better than baselines, and appendix results in other environments show the method to be better or same as baselines in the other environments.
- The proposed replanning method is general to work across environments, as evident by how it is implemented and by the experiments in multiple environments.
- Part of the code and the LLM prompts/examples included in the appendix

**Weaknesses:**

- Limited by the set of controllers and their abilities. Authors note that this is a limitation of the current work and could be addressed in future work.
- Some of the parts (e.g., success detector?) seem to rely on direct knowledge from the game. This may limit the application of the method to different environments.
- The "intelligence" of the agent stems from LLMs, and the result seems to rely on the quality of the LLM. This creates a hard dependency for using this method (you _need_ to use a high-end LLM, which might be costly and/or closed-source). Especially in this work, the core models were closed-source OpenAI models, which may come unavailable in future and/or change, rendering reproduction of these results impossible.
- Minor, but more analysis on the failure cases (e.g., when prompts fail, when they work, what seems to be problematic) would have provided more insight on how these types of planning should be done.

**Questions:**

1) How were the controller policies trained, exactly? Appendix mentions they are trained with imitation learning p(a | s, g), but where is the data collected? Looking at the shared code in "controller.py", there seems to be plenty of hardcoded actions (e.g., "smelt_w_furnace"). However, I am not sure if these functions are not used. Could you clarify how the controllers were trained, exactly?
2) What "offline trajectories" the selector model was trained on (Line 194)? If the trajectories were generated by an optimal policy that reaches goals as fast as possible, then the training might make sense, but if they are suboptimal, selector may learn a wrong sense of how fast player can reach objects. Seems like this is related to my first question.
3) Line 230: why different baselines/methods use different OpenAI models? I assume different baselines call for different type of generation (e.g., code vs. Text), but this is unclear. Please clarify/specify why same LLM was not used across planners.
4) Table 1: add explanation for what is the "+/-" metric. Interestingly, this is very high for the easy task but not for the hard tasks. Can you explain why?

### Comments (not questions)

- Use "DEPS" (and "DEP") acronym consistently after first defining it (defined again in line 312)
- Not a question/requirement, but seeing results with any openly shared LLM (i.e., weights are available for download and runnign locally) would be interesting to see.
- Another curiosity and not a requirement, but it would be great to see the model in action to see how well they behave. This is not a request, and I realize this is not a very objective measure, but it could give some intuition on how "directed" the agents are, for example (do they wonder around before doing the task, or do they directly go for the task?)
- Appendix line 501: sentence cuts off.
- Appendix Listing  6: first prompt is duplicated
- Figure 2 uses term "HPM", but this is not explained in the figure of caption. Same for VLM (which is defined later). For readibility, explain these acronyms in the caption or the figure.
- Line 224 paragraph: there should be a space between the method name and its acronym.
- Line 289: Citatations at the beginning of the sentence are bit out of place. Did you mean to use "citet" or "citep" commands instead?
- Line 291: VPT had higher diamond obtaining rates after RL-finetuning (I believe 2.5% was for diamond pickaxe). Double-check what numbers you want to compare against
- Line 292: I'd note here that DreamerV3 used a simplified environment where agent instantly destroyed blocks, effectively making the task easier than what the paper tackles.
- Line 338: A nitpick, but I would recommend rephrasing the final sentence to indicate you are planning to work on this but also indicate that this work lays solid foundation for anyone to build on.

**Limitations:**

Authors note the limitations. Especially noteworthy is the mentioned limitation of using closed-source models, and the suggestion to replicate the results with open source models.

Authors did not include broader impact/societal impact section. This is not required for this paper (no immediate impact), however I'd recommend adding one regardless so that authors think of the implications. E.g., these are very capable models, but if they solve tasks in misaligned/harmful ways, they may end up doing harm if ever deployed. We should aim to understand this alignment and/or analyze how model approaches the tasks.


## Rebuttal acknowledgement

I have read authors' rebuttal and new results which did address my concerns, and I updated my score from 8 to 9 (before discussion period closed).

---

> ### Author Rebuttal · Authors · 2023-08-10
>
> Thanks for your comments. We provide clarifications below on the major comments, which will be included in the final version.
> > Concern on the set of controllers and their abilities.
> >
>
> DEPS primarily focuses on generating more feasible plans for long-horizon tasks in an open-world scenario, using a pretrained language-conditioned controller. The training techniques can be found in Cai et al.. In our future work, we aim to scale up the language skill set of the controller to enable DEPS to accomplish a wider range of tasks.
>
> > Concern on relying on knowledge of game and applications to other environments.
> >
>
> The goals we plan usually cause changes in the items in the agent's inventory, so the success detector gives the execution result by observing the item information in the inventory. The success detector can also use LLM for automatic determination. We discovered that the Language Model (LM) trained on a large-scale language corpus from the internet demonstrates strong generalization capabilities across different environments. As a result, we employ this LM as a zero-shot planner, referred to as LLM. Minecraft's open-world nature makes it challenging to summarize its vast internet-scale knowledge using formal languages like PDDL. Hence, utilizing a pretrained LM is an excellent option. Furthermore, we conducted experiments in ALFWorld and Tabletop Manipulation environments to assess DEPS's generalization abilities independently. The results can be found in Appendix Section A.1 and Section A. 2.
>
> > Concern on Language Models.
> >
>
> We conduct experiments on open-sourced LLaMA2. Please refer to Q1-A1 and Table 1 in the global rebuttal PDF.
>
> > More analysis on the failure cases.
> >
>
> Thank you for the comment. We commit to including more discussions on failure modes in the final version. Here, we provide some analysis for your reference in Q3 and A3 in global rebuttal region.
>
> > Concern on controller training.
> >
>
> The controller is composed of two parts: the policy obtained from Behavior Cloning training, which includes mining items, and the policy completed using shell, which involves crafting and smelting items. We utilized the method described in Cai et al. to gather offline goal-conditioned data and train a goal-conditioned behavior cloning policy. To be specific, we collected data by running the VPT agent in Minedojo and labeling its trajectory as language goals based on environmental information. For further information, please refer to Cai et al.'s work. We will commit to including more details to make our paper self-contained.
>
> > Concerns on training data for selector.
> >
>
> The selector is trained jointly with the controller, using the same offline data as the controller. Specifically, the predicted horizon information by the selector is used as an auxiliary loss to assist. During actual training, in order to reduce errors caused by possible suboptimal results, we segmented the horizon values and optimized them using classification loss. Finally, the horizon prediction loss of the selector was reduced to 0.15 (as a comparison, the final language goal-conditioned action loss was 1.8).
>
> > Concern on OpenAI models in baselines.
> >
>
> The choice of different language models is related to the prompt template. Huang et al and Chain-of-Thought algorithm tend to use text description more, so we used GPT-3 (text-davinci-03), while Code as Policies and ProgPrompt tend to use code as a prompt template, so we used Codex (code-davinci-02). Since the baselines we compared did not experiment in Minecraft, we reproduced their algorithms in the Minecraft environment (we described the prompt templates of different methods in detail in Appendix Section-G.1.2). To ensure consistency with the original algorithms, we used language models that were also used in the paper for different algorithms (e.g., GPT-3 for Huang et al and Codex for ProgPrompt).
>
> > Explanation for "+/-" metric in Table 1.
> >
>
> +/-  denotes standard deviation (std) of success rate across different tasks within the same task group. The absolute magnitude of variance does not matter, if you take a look at the ratio, hard tasks still have a relatively higher variance. In addition, the execution of all tasks is under random seed and random initial location, which may cause significant variance.
>
> > Results on open-source LLM.
> >
>
> The experiment results are shown in Table 1 of the global rebuttal PDF.
>
> > Model in action.
> >
>
> We have added screen recordings(gifs) of the agent performing tasks such as ObtainPainting, ObtainStoneStairs, and ObtainWoodenSlab in `Supplementary/Code/imgs` folder, please feel free to take a look. Thank you for your suggestion, we will add more visualizations to the paper.
>
> > Comments on typos.
> >
>
> Thanks for noting. All fixed.
>
> Cai et al. Open-world multi-task control through goal-aware representation learning and adaptive horizon prediction. CVPR 2023.

---

> > ### Comment · Reviewer_FjP3 · 2023-08-14
> >
> > Thank you for your extensive reply and additional experiments! The results with LLaMA2 and DEPS improvement over any vanilla-planning LLM are especially encouraging.
> >
> > I have increased my score from 8 to 9 given these additional experiments and explanations in other comments. This work includes  novel-enough methods that both show strong performance _and_ are more available than just "one big model" (the results show that this method works very well with any LLM, and underlying LLM does not affect the result as much). The work also includes extensive benchmarks regarding various planning methods, underlying controllers and the LLMs. Finally, being done in Minecraft, the experiment pipeline is available to anyone, unlike robotics which are prohibitively expensive for many many labs.
> >
> > Overall, I believe this work is a good exemplar of how the new capabilities of LLMs should be pushed and tested. Other reviewers pointed weaknesses like number of seeds. I do agree the 30 seeds may not be enough, however the improvement is visible in terms of maximum goal methods reached. Reaching later goals is very difficult in Minecraft, and there is a clear gap-improvement if one model is able to reach it even once vs. one that never reaches it.
> >
> > I have only two requests for the authors for camera ready:
> > * Please add the open-source model results (with potentially more seeds, if possible) to the main paper, or at least appendix.
> > * Open-source the code, and if possible, the controller models you used. I realize this can be a huge effort, but this is what enables others to build on your work and push the limits, much like your work was enabled by the open source VPT model and MineDojo framework.

---

> > > ### Author Response · Authors · 2023-08-14
> > >
> > > Thanks for your very encouraging reply! We're happy to see our rebuttal has addressed your concern. We will commit to including what you have suggested: 1) results with open-sourced models; 2) open source the code and controller model, in the final version.

---

### Official Review · Reviewer_LCb8 · 2023-07-07

**Soundness:** 3 good
**Presentation:** 3 good
**Contribution:** 3 good
**Rating:** 5
**Confidence:** 3

**Summary:**

The paper presents a novel approach called "Describe, Explain, Plan and Select" (DEPS) for task planning of multi-task embodied agents within open-world environments. The authors identify two main challenges in this area: the need for accurate and multi-step reasoning due to the long-term nature of tasks, and the inefficiency or infeasibility of plans created by vanilla planners that do not consider the ease of achieving sub-tasks when ordering parallel sub-goals.
DEPS addresses these challenges by integrating a description of the plan execution process and providing self-explanation of feedback when encountering failures during the extended planning phases. It also includes a goal selector, which ranks parallel candidate sub-goals based on the estimated steps of completion, refining the initial plan.
The authors conducted experiments in the open-world game Minecraft, marking the first zero-shot multi-task agent that can robustly accomplish 70+ tasks, nearly doubling the overall performances. The method also proved effective in non-open-ended domains such as ALFWorld and tabletop manipulation. The authors concluded with ablation and exploratory studies detailing how DEPS outperforms counterparts and provides a promising update on the ObtainDiamond grand challenge.

**Strengths:**

- The overall pipeline is useful. The paper introduces a novel planning approach, DEPS, that integrates description of the plan execution process and self-explanation of feedback when encountering failures.
- The fact that DEPS is capable of zero-shot learning, meaning it can generalize to tasks it has not seen during training, is a strength. This makes it a flexible and powerful tool for task planning in open-world environments. However, this is not an entirely new concept, as previously shown in e.g., Code as Policies (Liang et al. 2022), Text2Motion (Lin et al. 2023), ProgPrompt (Singh et al. 2022), Inner Monologue (Huang et al. 2022), Socratic Models (Zeng et al. 2022).
- Beyond that, the proposed use of a goal selector that ranks parallel candidate sub-goals based on the estimated steps of completion is a strength. It allows for more efficient and feasible planning and is verified empirically.
- The performance seems good. The paper also includes detailed ablation and exploratory studies, providing a thorough understanding of why DEPS outperforms other methods. I will include more discussion below.
- Additional details in appendix and source code are provided, which are useful to ensure the reproducibility of the paper and good practice for ML community.

**Weaknesses:**

- Although the overall pipeline sounds promising, there has been several similar work in zero-shot task planning on Minecraft and other embodied environments: e.g., Code as Policies (Liang et al. 2022), Text2Motion (Lin et al. 2023), ProgPrompt (Singh et al. 2022), Inner Monologue (Huang et al. 2022), Socratic Models (Zeng et al. 2022).
- However, the selector is a completely new module and distinguish the work from others, while I feel it could be highlighted more. I list some points in “Questions”.
- The explanation of the selector could be improved. This seems like a well motivated module to have and plays a key role. But, for example, what does this module take as input? More concretely, how are plan $P_t$, agent full state $s_t$, and goal $g_t$ represented? How does it train and what is data collection procedure? More details could be provided or pointed to appendix (not seen).
- The empirical study is great and supports the claims in general. But I have some questions/comments and list them in “Questions”.
- Lack of Discussion on Failure Cases: While the paper mentions that DEPS provides self-explanation of feedback when encountering failures, it doesn't delve into specific instances where DEPS failed and how these failures were addressed.
- Another general issue, dependency on LLMs: DEPS relies heavily on specific LLMs, which may limit its applicability in environments where such models are not feasible due to computational or data constraints, or domain knowledge not available as in Minecraft.

**Questions:**

- L122 — Intuitively, the selector seems useful. But how generalizable is this selector to unseen goals/plans? Can it always produce helpful output for selecting plans?
- L241, “This validates that “describe, explain and plan” can estimate the reason for current plan failure and correct the original ﬂawed plans.” — Why this is true? Is there any more concrete example? Why the improvement necessarily mean that “DEP can estimate reason for failure”?
- L245 — The success rate boost from selector sounds promising. I wonder if there is any other measure that can reflect the performance gain, such as average steps?

**Limitations:**

Yes

---

> ### Author Rebuttal · Authors · 2023-08-10
>
> Thanks for your comments. We provide clarifications below on the major comments, which will be included in the final version.
>
> > Similar works in zero-shot task planning.
> >
>
> Please refer to Q2 and A2 in the global rebuttal.
>
> > The explanation of the selector could be improved. This seems like a well motivated module to have and plays a key role. But, for example, what does this module take as input? More concretely, how are plan P_t, agent full state s_t, and goal  g_t represented? How does it train and what is data collection procedure? More details could be provided or pointed to appendix (not seen).
> >
>
> We will include more details on the selector in the final version. Below, we elaborate more on it for your reference:
>
> 1. high-level overview of the selector (sec. 3.3):
>
> The Selector is a vision language model that takes the language goal (e.g., obtain_oak_log, obtain_iron_ore) and the agent’s current visual observation at time t as input, and predicts the number of steps to achieve the language goal from now (time t). We may use this prediction to compare the proximity between the agent and multiple language goal candidates, then select the language goal that requires the least number of steps to complete.
>
> 2. representation
>
> P_t is a sequence of language goals, each language goal g_t is a natural language sentence, e.g. “mine 1 oak log”, agent full state s_t is an image (visual observation).
>
> 3. training and data collection
>
> As mentioned in sec. 3.3, we jointly train the goal-conditioned controller and the selector with  <goal, trajectory demonstration> pairs. For each training trajectory, the selector is trained to map any state (e.g. at time t) and the goal to the distance T-t between that state and the termination state (at time T). Please note that we adopt this horizon predictive technique from Cai et al. and we will include more details in the final version to make our paper self-contained.
>
> Cai et al. Open-world multi-task control through goal-aware representation learning and adaptive horizon prediction. CVPR 2023.
>
> > Discussion on Failure Cases.
> >
>
> Thank you for the comment. We commit to including more discussions on failure modes in the final version. Here, we provide some analysis for your reference in Q3 and A3 in the global rebuttal region.
>
> > Dependency on LLMs
> >
>
> We include results of DEPS based on different LLMs: Codex (Table-1), ChatGPT, GPT-3, and GPT-4 (Table-11 in the appendix). It can be seen that the difference in performance among them is not significant, proving that DEPS does not rely heavily on specific LLMs. We also conduct experiments on DEPS with open-sourced LLaMA2-70B and finetuned smaller open-sourced LLaMA2-13B(using the data collected from internet). The results are shown in Table 1 in global rebuttal PDF. Please refer to Q1-A1 in the global rebuttal section for more discussion.
>
> > L122, Intuitively, the selector seems useful. But how generalizable is this selector to unseen goals/plans? Can it always produce helpful output for selecting plans?
> >
>
> Thanks for your comment. Investigating whether our horizon-predictive selector can generalize to unseen goals/plans is certainly an exciting future extension of the current work. Also, please note that we already explored a CLIP-based selector, which should be able to generalize to unseen goals/plans, but its overall performance is not on par with the horizon-predictive selector.
>
> > L241, “This validates that “describe, explain and plan” can estimate the reason for current plan failure and correct the original ﬂawed plans.” — Why this is true? Is there any more concrete example? Why the improvement necessarily mean that “DEP can estimate reason for failure”?
> >
>
> Thank you.  Here is an additional ablation of DEP but without the explainer: on 71 Minecraft tasks, the success rate has dropped from 39.36 (DEP) to 30.25 (DEP w/o Explainer). We will include this in the final version. As for concrete examples of how DEP can estimate the reason for failure, please refer to the Q3 and A3 in the global rebuttal region.
>
> > L245, The success rate boost from selector sounds promising. I wonder if there is any other measure that can reflect the performance gain, such as average steps?
> >
>
> Our best horizon-predictive selector (HPM) indeed leads to a less averaged number of steps of task completion compared to other implementations (eg. Random, CLIP, and MineCLIP) and not using the selector at all (Fixed). The average step results are shown in Table 3 in the global rebuttal PDF.

---

> > ### Author Response · Authors · 2023-08-16
> >
> > Dear reviewer, it has been a few days since the discussion period began, and I am curious whether our response has addressed some of your concerns. If you have any further questions, please feel free to discuss them with us.

---

### Official Review · Reviewer_Qeax · 2023-07-26

**Soundness:** 4 excellent
**Presentation:** 4 excellent
**Contribution:** 3 good
**Rating:** 6
**Confidence:** 3

**Summary:**

In order to solve the long-term planning problem in the open world, this paper designed an interactive planning framework based on LLMs, and trained the first planning-based agent that successfully implements the ObtainDiamond task in the Minecraft scene.


**Strengths:**

1. The article is clear and the idea proposed by the author is easy to follow.

2. The work is solid, and the method has been verified to be effective on multiple benchmarks. Ablation experiments comprehensively analyze the necessity of each module.


**Weaknesses:**

1. The failure criteria of the controller that triggers the re-planning mechanism is not clear enough. Can the author explain in detail how to determine whether the failure is caused by the planning errors generated by the planner or the controller executing actions incorrectly?


**Questions:**

Please refer to the weakness above.

**Limitations:**

Yes.

---

> ### Author Rebuttal · Authors · 2023-08-09
>
> Thanks for your comments. We provide clarifications below on the major comments, which will be included in the final version.
> > The failure criteria of the controller that triggers the re-planning mechanism is not clear enough. Can the author explain in detail how to determine whether the failure is caused by the planning errors generated by the planner or the controller executing actions incorrectly?
>
> Re-planning is triggered whenever the controller fails to complete the designated task within a predefined number of environment steps, regardless of whether the failure is caused by the planning error or the controller’s error. The explainer in our DEPS model can handle both aforementioned failure modes. Specifically, DEPS will self-correct using the error reasons provided by its explainer if the explainer identifies errors in the plan. However, if no errors are identified, DEPS will assume the controller’s error and re-execute the original plan.

---

> > ### Author Response · Authors · 2023-08-16
> >
> > Dear reviewer, it has been a few days since the discussion period began, and I am curious whether our response has addressed some of your concerns. If you have any further questions, please feel free to discuss them with us.

---

> > > ### Comment · Reviewer_Qeax · 2023-08-19
> > >
> > > The rebuttal has addressed my concern, and it is not a very crucial one. I am happy to keep voting for acceptance of this paper.

---

### Author Rebuttal · Authors · 2023-08-10

Appreciate the reviewers’ constructive comments. We provide clarifications on the common comments as follows, which will be amended in the final version.

> Q1: Concerns on using OpenAI closed Language Models.

A1: We noticed that there are some concerns about DEPS using OpenAI's language model, and the performance of DEPS highly relies on high-quality Language Models. Vanilla LLM cannot solve the problem of planning in open-ended environments (see Figure-1 in Introduction), so we designed a pipeline called "describe, explain, plan and select" to generate feasible plans in open-ended environments.

We presented the performance of DEPS using Codex in Table-1. To assess DEPS's compatibility with various language models, we also included its performance with different language models like ChatGPT, GPT-3, and GPT-4 in Table-11 in the Appendix. The table reveals notable variations in Vanilla Planner's performance depending on the language model used. However, after improving DEPS, there were no significant differences observed among the performances of different language models. Our experiments demonstrated that distinct language models offered reliable planning performance for Minecraft tasks within specific groups.

Since we use pretrained LLM as a planner, it indeed requires exposure to a large amount of Minecraft-related corpus during the pretraining phase. Considering that Minecraft is one of the most popular games worldwide, there is relatively abundant data about Minecraft available online. We conducted experiments using open-source pretrained LLaMA2-70B on several Minecraft tasks and found that DEPS based on LLaMA2 also performs reliable planning under Minecraft conditions. Considering limited training data used by LLaMA2, we further finetuned an open-source language model (LLaMA2-13B) using Minecraft texts obtained from the internet which exhibited better planning performance. The results are shown in Table 1 in global rebuttal PDF. So we don't think this will become a bottleneck in the future.

> Q2: Difference with other similar works in zero-shot task planning, including Inner Monologue, Code as Policies, etc.

A2: Although these models achieved good planning performance in various benchmarks, most of them suffer from the challenges brought by open-world domains such as Minecraft, and DEPS proposes substantial improvement over these approaches to address the open-world planning challenges.

First of all, as illustrated in the Introduction Section, we identified two main challenges in the open-world domains which would cause failure for the above methods: 1) executing plans within an open-world environment like Minecraft requires accurate and multi-step reasoning due to the long-term nature of tasks, and 2) vanilla planners do not consider the ease with which the current agent can accomplish a given sub-task when ordering parallel sub-goals within a complex plan, resulting in inefficiency or even failures. In the related works section, we have shown the differences between DEPS and these methods and argue that our newly introduced **interactive planning** (through describe, explain, plan, and select)in DEPS is the key to addressing these challenges.

Further, we have established baselines including  Code as Policies, ProgPrompt, and Inner Monologue (Text2Motion is a concurrent work) in our experiments with minimum necessary modifications for them to be applied to the Minecraft domain(the modification details of the prompt can be found in Appendix Section G.1.2). The experiment results in Table-1 show that DEPS is more effective in open-world Minecraft environments than these baselines, thanks to our substantial improvement, i.e. the interactive planning techniques.

> Q3: More analysis on the failure cases.

A3: There are two main types of failures: 1) the plan produced by DEPS is wrong, which can be fixed by the explainer. 2) the plan is correct, but the low-level goal-conditioned controller somehow failed to execute it. We will provide some examples for them below:

1. The plan is wrong about crafting materials:

    In this example, the plan includes crafting 8 cobblestones into a furnace. The explainer found that there are not enough cobblestones in the inventory. Therefore, the explainer asks the planner to modify the plan by first mining enough cobblestones and then crafting them into a furnace.

    ```
    Failed Action: craft({'furnace':1}, {'cobblestone':8}, 'crafting_table'); # step 3: craft 1 furnace from 8 cobblestone on crafting_table
    Description: Now I locates in Forest biome. My inventory has 2 log, 1 crafting_table, 1 wooden_pickaxe, 6 cobblestone.
    Explanation: Because crafting furnace needs to have at least 8 cobblestone in inventory, but my inventory only has 6 cobblestone, which is not enough for crafting furnace. So I need to mine 8 cobblestone first.
    ```

2. The plan is wrong about tools:

    In this example, the plan includes mining iron_ore. The explainer found that the agent does not have stone_pickaxe, which is required for this. Therefore, the explainer asks the planner to modify the plan by crafting a stone_pickaxe first.

    ```
    Failed Action: mine({'iron_ore':1}, null); # step 5: mine 1 iron_ore without tool
    Current Inventory: null
    Explanation: Because mining iron_ore needs to use the tool stone_pickaxe, but my inventory does not have stone_pickaxe. So I need to craft stone_pickaxe first.
    ```

3. The low-level controller failed:

    In this case, the explainer will find no error in the plan, therefore DEPS will simply re-execute the plan.


We visualized some explanations of the explainer output in Figure-2. The Explainer prompt and some examples are placed in Listing 1 in Appendix C.3. We will also add more details about the explainer output in the paper.

---

### Decision · Program_Chairs · 2023-09-21

**Decision:**

Accept (poster)

**Comment:**

The authors present DEPS, an approach towards planning for embodied agents. DEPS uses feedback of the scene and errors in the LLM-planning processes to select subgoals. This is demonstrated on a variety of Minecraft tasks to perform even more complex tasks.

The reviewers generally all agreed the work should be accepted, and I concur. For the cameray ready, the authors should open source code and models as well as list additional information discussed in the rebuttal.